# Assessing the impact of artistic and cultural activities on the health and well-being of forcibly displaced people using participatory action research

Clelia Clini,[1] Linda J M Thomson,[2] Helen J Chatterjee[2]

¹Institute for Media and Creative Industries, Loughborough University London, London, UK
²Genetics, Evolution and Environment, UCL Biosciences, University College London, London, UK

**Correspondence to**
Professor Helen J Chatterjee;
h.chatterjee@ucl.ac.uk

## ABSTRACT

**Objective** Drawing on a growing body of research suggesting that taking part in artistic and cultural activities benefits health and well-being, the objective was to develop a participatory action research (PAR) method for assessing the impact of arts interventions on forcibly displaced people, and identify themes concerning perceived benefits of such programmes.

**Design** A collaborative study following PAR principles of observation, focus groups and in-depth semistructured interviews.

**Setting** London-based charity working with asylum seekers and refugees.

**Participants** An opportunity sample (n=31; 6 males) participated in focus groups comprising refugees/asylum seekers (n=12; 2 males), volunteers (n=4; 1 males) and charity staff (n=15; 3 males). A subset of these (n=17; 3 males) participated in interviews comprising refugees/asylum seekers (n=7; 1 males), volunteers (n=7; 1 males) and charity staff (n=3; 1 males).

**Results** Focus group findings showed that participants articulated the impact of creative activities around three main themes: skills, social engagement and personal emotions that were explored during in-depth interviews. Thematic analysis of interviews was conducted in NVivo 11 and findings showed that artistic and cultural activities impacted positively by helping participants find a voice, create support networks and learn practical skills useful in the labour market.

**Conclusions** The study expanded on arts and well-being research by exploring effects of cultural and creative activities on the psychosocial well-being of refugees and asylum seekers. By focusing on the relationship between arts, well-being and forced displacement, the study was instrumental in actively trying to change the narrative surrounding refugees and asylum seekers, often depicted in negative terms in the public sphere.

## INTRODUCTION

The recent All-Party Parliamentary Group on Arts Health and Well-being Inquiry Report shows a growing acknowledgement of the impact of arts and creative practices, stating 'arts engagement has a beneficial effect upon health and wellbeing and therefore has a

### Strengths and limitations of this study

► Focus on collaborative research.
► Different cultural and artistic activities studied.
► Researcher spent 5 months in research setting.
► Greater number of female than male participants.
► Focus groups attracted relatively low numbers of refugees/asylum seekers.

vital part to play in the public health arena' (APPGAHW, p11).[1] Consensus among practitioners defines well-being as 'the dynamic process that gives people a sense of how their lives are going through the interaction between their circumstances, activities and psychological resources or mental capital' (NEF, p3).[2] Rather than depending on a single factor, well-being is seen as the result of interplay between different elements, distinguishing personal well-being (positive emotion, life satisfaction, vitality, resilience and self-esteem) from social well-being (supportive relationships, trust and belonging).[3] Other authors emphasise 'resilience and flourishing, rather than just surviving' (Ander *et al*, p243),[4] a crucial point when investigating the situation of forcibly displaced people who often survive in extreme conditions and whose experience is often discussed in terms of dehumanisation.[5 6]

Research conducted in the field of forced displacement and mental health shows that 'asylum seekers and displaced people report high rates of pre-migration trauma' (Vaughan-Williams, p275),[6] and typically display significant levels of depression, anxiety, posttraumatic stress disorder (PTSD) and non-affective psychoses.[7–11] Refugees and asylum seekers often encounter postmigration living difficulties, such as socioeconomic disadvantage, employment barriers, social and emotional isolation, racism and hostility,

**BMJ**

experience of detention and uncertainty related to the asylum application process.[7–9 12] If, as authors suggest,[13–17] there is a strong link between mental health and socioeconomic conditions in terms of employability, income and housing, then the mental health of unemployed asylum seekers living on low incomes in poor quality housing is likely to be worse than that of the general population.

Arts-and-health practitioners believe that 'aesthetics act upon our senses to make us feel more, hear more and see more than we otherwise might… feelings are intertwined with mental, physical, spiritual and social health' (Prior, p4).[18] Several studies explored the relationship between arts, health and well-being and a growing body of evidence suggests that cultural participation (including music, art making, theatre, dance, museum and heritage activities) enhances human health and well-being.[1 19–27] Research conducted in museums-and-health, for example, suggests that 'museum and art gallery encounters can help with a range of health issues, enhance wellbeing, and build social capital and resilience' (Chatterjee and Noble, p286).[27] According to research conducted in Norway with over 50 000 adults,[28] participation in both receptive/passive and creative/active cultural activities was significantly associated with good health, life satisfaction, and low anxiety and depression. Findings demonstrated how manual creative practices such as knitting had a positive impact on people suffering from depression and posttraumatic stress disorder because 'the movements involved in knitting are bilateral, rhythmic, repetitive, and automatic' (Cuypers *et al*, p40).[28] The authors attributed the positive mood of knitters to enhanced production of serotonin resulting from repeated movements, and that bilateral processes appeared to engage brain capacity and facilitate a meditative-like state more readily than unilateral ones. Although creative activities are seen to provide major benefits for refugees and asylum seekers, researchers have become 'increasingly conscious of the value of recording and analysing what has been happening' (Robjant *et al*, p1).[8]

The current study was conducted in the light of mental health and arts-in-health/museums-in-health evidence, and the large number of organisations that engage migrants, refugees and asylum seekers in the arts, with nearly 200 of these in the UK.[29] The current research was conducted at the Helen Bamber Foundation (HBF), a charity offering support to refugees and asylum seekers as victims of torture and human rights violations. HBF adopts a holistic approach to support its clients that includes cultural activities within a three-phase model of integrated care comprising stabilisation (medical and legal support), intervention (trauma-focussed therapy and general psychological care) and integration (attending the creative, computing and English classes, and interacting with other clients). The model is based on the needs of refugee/asylum seekers who often require assistance on several fronts (eg, psychological support might not be effective if a person does not have a place to sleep or lives in a dangerous situation, though

giving people a place to sleep is not enough to overcome trauma). Their creative arts programme (CAP) led by volunteers includes art (painting and drawing), photography, textiles (dressmaking and knitting) and singing, and is attended by c.100 clients. The study took the view that arts participation would enhance the sense of well-being by allowing participants to form meaningful relationships based on trust and mutual support (improving social well-being) and enhance self-esteem and self-confidence (increasing personal well-being).

To provide voice and agency for participants, the study used a collaborative approach to explore client experiences of creative activities with HBF clients, staff and volunteers acting as coresearchers. The research was operationalised according to the principles of participatory action research (PAR)[30 31] as a 'way of opening up space for dialogue and conversation' (Nicolaidis and Raymaker, p28),[32] aiming to 'understand and also improve a particular situation',[32] both pertinent aspects when working with displaced people. The PAR approach encourages an active contribution in the production of knowledge within a collaborative framework, emphasising 'equal partnerships' (Daykin and Stickley, p167),[33] and the 'role of the participant in the design, implementation, and dissemination of the research' (Vaughn and Jacquez, p78).[34] The collaborative character of PAR is grounded in efforts to 'democratise the research process' (Blumenthal, p3),[35] so 'equal weight and consideration is given to the contributions of both the community and academic partners' (Braun and Clarke, p170)[36] again valuable for everyone involved but specifically refugees/asylum seekers. One of the pillars of PAR, especially when community-based is ownership.[37]

By offering equal weight to client contributions focusing on the relationship between forced displacement and arts participation, the study aimed to empower participants by developing their sense of ownership in the project. Through collaborative working between the charity, displaced people and academics, the purpose of the research was to codevelop a methodological approach to address the needs of refugees/asylum seekers and to coproduce interview questions that could be applied in the UK and international settings. The objectives were to expand on arts-in-health and collaborative research approaches by exploring the benefits of engagement in cultural and creative activities on the health and well-being of forcibly displaced people and contribute to the current debate on migration and public health.

## METHOD
### Design
Qualitative research conducted comprised observation, focus groups and one-to-one semistructured interviews with clients, volunteers and staff as coresearchers within a collaborative PAR approach.[38]

## Participants

The study recruited an opportunity sample of volunteer participants through information leaflets and contact details in reception. Clients were approached by the researcher using two recruitment criteria: (i) clients had received therapy for at least 2 years prior to the research and had entered the integration stage of the model of integrated care; this meant that though still vulnerable, clients had established a therapeutic relationship of trust enabling them to form safe relationships in the wider community and move forward with their lives; and (ii) they had attended at least one of four CAP groups (art, photography, singing and textiles) for around 2 years. All participants spoke English sufficiently well to participate having attended English classes for about 2 years as part of integrated care. In total, 31 (6 males) participants volunteered for the study and attended focus group comprising 12 (2 males) refugees/asylum seekers, 4 (1 males) volunteers and 15 (3 males) charity staff. Of these, 17 (3 males) participated in interviews comprising 7 (1 males) refugees/asylum seekers, 7 (1 males) volunteers and 3 (1 males) staff.

### Patient and public involvement

HBF staff were instrumental in the application for research funding and collaborated with the researcher to determine the study's objectives. HBF clients were involved in the recruitment process by recommending that other clients join them in the research, though the researcher ensured that they matched recruitment criteria (above). In keeping with PAR, research questions and outcome measures concerning the effects of cultural and creative activities on the psychosocial well-being of refugees/asylum seekers were determined by coresearchers (HFB clients, volunteers and staff) with differing involvement depending on their role at the charity. Focus groups and interviews gave voice to clients' priorities and preferences but, due to their vulnerability, no personal information was requested as the process of remembering could have been difficult.[7 10] Although some participants mentioned personal issues and experiences, they were not encouraged to do so. Data generated by coresearcher involvement in the research collaboration informed focus groups, in turn focus group outcomes led to the development of questions for the in-depth interviews to follow. In addition to participation in focus groups and interviews, a core group of seven coresearchers (three clients, two volunteers, two staff, with one male per category) were involved in design of the interview guides, discussion of emerging themes and dissemination of the research findings to other participants for verification and comments, in accordance with PAR principles.[33] The results of the research were fed back to everyone involved with HBF through ongoing communication with the researcher, an internal report lodged at the charity, and an end-of-project event.

### Data collection

Data collection carried out by the location-based researcher consisted of three stages: stage 1 (months

> ### Box 1 Focus group questions
>
> **Clients**
> 1. Why do you like the Helen Bamber Foundation (HBF) creative arts programme?
> 2. How would you gather information or evidence about the benefits of artistic activities?
> 3. Which creative arts classes have a positive impact on the lives of refugees?
> Circle the option(s) that you think work best or add another one.
> Arts and Crafts Drawing Films Knitting Photography Singing Textiles.
>
> **Volunteers**
> 1. Why do you volunteer at the HBF?
> 2. How do you think your group benefits clients?
> 3. How would you collect evidence on the effects of participating in artistic activities?
>
> **Staff**
> 1. Why do you recommend clients to attend creative arts groups?
> 2. What do you think are the benefits of attending arts groups?
> 3. How would you collect evidence on the effects of participating in artistic activities?

1–2) participant observation; stage 2 (months 3–4) focus groups and stage 3 (month 5) semistructured interviews. In stage 1, participant observation was used as a starting point 'for studying how organisations work, the roles played by different staff and the interaction between staff and clients' (Pope *et al*, p32).[38] The researcher attended creative classes for clients and spent a day a week working from HBF (12–15 hours per week) to determine the nature of focus groups. In stage 2, four focus groups were held, first and final with clients (n=8, n=4, respectively), second with volunteers (n=4), and third with staff (n=15). In the focus groups, participants were asked to discuss research questions (box 1) involving ways of gathering information on the impact of creative activities, and reasons for attending groups (or in the case of volunteers and staff, reasons for involvement). All focus groups were organised informally to facilitate development of discussion among participants and allow an exchange of experiences and ideas. Ideas that emerged during focus groups laid the basis for first drafts of interview questions, tailored to clients, volunteers or staff (box 2). In line with the collaborative ethos, drafts were circulated to receive feedback from coresearchers prior to the interview stage, and volunteers and staff were consulted and asked to provide comments and suggestions for interview questions. For stage 3, clients (n=7), volunteers (n=7) and staff (n=3) participated in interviews with the researcher using a semistructured format to allow free expression within the research constraints and limit potential research bias.[30 39]

### Data analysis

A first informal thematic analysis was conducted by the researcher while writing field notes consisting of a chronicle of descriptive rather than analytic events both observed and provided by coresearchers as the 'raw material of the research' (Kemmis *et al*, p38).[30] The process

## Box 2 Interview questions

**Clients**
1. How long have you been attending the creative arts programme (CAP) classes for?
2. Which groups do you attend?
3. Why did you choose this/these groups(s)?
4. What do you like the most about these groups?
5. Is there anything about the groups that you find difficult?
6. Anything you would like to change?
7. Is there a group you prefer? Which one? Why?
8. How do you feel when you attend this group?
9. Do you think there are benefits of attending these groups? If so what are they?
10. During our previous meeting, it emerged that participants feel that groups allow them to learn new or improve already acquired skills. Do you agree? Have you learnt any new skill? And what do you think about the possibility of learning new skills? (How does that impact your life?)
11. Another element which emerged during our meeting is that people enjoy the social aspect of these groups: meeting people and finding new friends. What do you think about this? [supporting questions: do you like being around other people? Why? What happens when you attend a session? Have you made new friends? How do you feel about that?]
12. Among all of the reasons why you enjoy attending the creative arts group, which one is the one that you feel strongest about?
13. Why do you keep attending?
14. If you could compare the way you were feeling before joining any of these activities and the way you feel now, would you say you feel any different? Explain.
15. Creative activities of course are part of the support that Helen Bamber Foundation (HBF) offers to its clients. Do you think that you would feel the same about yourself today even without attending these creative activities groups? Why?
16. Do you think that taking part in these creative activities has had any influence on the way you see yourself? And the way you see yourself in London?
17. If you were to recommend someone to join a group, what would you say?

**Volunteers**
1. What do you teach?
2. How long have you been volunteering for?
3. Why did you decide to volunteer?
4. How did you learn about the HBF?
5. What do you think your group offers clients?
6. What are the main challenges you face/have faced as a volunteer?
7. Do you find it difficult to engage clients in the activity you coordinate? What do you think clients like about the way you (and your colleagues) handle the sessions? Is there anything that clients do not like?
8. How would you describe a typical session?
9. What do you think people like *the most* about your group?
10. Do you ever discuss emotions with clients? Or their personal situation?
11. Look at the notion of well-being defined by NEF (New Economics Foundation):
*Well-being as 'the dynamic process that gives people a sense of how their lives are going through the interaction between their circumstances, activities and psychological resources or mental capital.'* (NEF 2008, p3)

## Box 2 Continued

According to the NEF, there are different components to well-being, such as:
'Personal wellbeing (emotional wellbeing, satisfying life, vitality, resilience, self-esteem and positive functioning) and social wellbeing, including supporting relationships and trust and belonging.' (NEF 2009)
12. Do you think that the activity you lead has any impact on the well-being of clients? Elaborate.
13. If you were asked to recruit people for your group, what would you say to convince them to join?
14. Would you use different words to promote your group with men and women? Why?
15. Do you think that your group makes a difference in the lives of clients? How? Why?
16. And what about your own life? What is the impact of your volunteer activity on your own life?

**Staff**
1. How long have you been working at HBF for?
2. Why did you decide to work here?
3. What is your role?
4. As part of your work, do you have any direct contact with clients?
5. Could you say a few words on your relationship with clients?
6. How would you describe your own experience at HBF so far?
7. What are the main challenges that you face/have faced in your position?
8. Do you think that your own gender influences the ways in which clients relate to you?
9. In your position, do you discuss clients' personal situations and feelings?
10. If so, do you think that the cultural background of clients affects the way in which you discuss emotions? If it does, how and why?
11. What are the main challenges of discussing emotions with clients?
12. Could you say a few words on the Model of Integrated Care? Do you think it works? And if so, why?
13. Do you recommend clients to attend any of the CAP groups? What do you say in this case?
14. What role do you think cultural and creative activities play in the recovery of clients?
15. Are there any particular activities that, in your opinion, are more popular with clients?
16. Would you say that the programme makes a difference in the lives of clients? If yes, please elaborate.
17. Have you witnessed clients benefiting from their own participation in the creative arts programmes? If yes, in what ways?

of transcribing notes was used to identify key points connected to the research question. This preliminary analysis informed focus group topics where points were discussed more in detail. After the first round of focus groups, all notes were compiled into a word document and repeatedly read by the researcher and coresearchers to search for recurring topics and themes. Thematic analysis of focus group outcomes were explored in detail during interviews. All interviews were recorded and transcribed to become familiar with the data and begin the coding process.[37] All details in the transcripts were recorded verbatim (eg, sighs, laughter, silences and tears) as their exclusion could have changed the meaning expressed.[30] Transcriptions were uploaded into NVivo11

to produce nodes from coding-relevant concepts. NVivo's word frequency query facilitated the search for keywords (including stem words and synonyms) grouped under the same theme. The text search query provided a comprehensive analysis of data. After coding of data into initial themes,[37] a review of themes followed to refine the analysis.

## RESULTS
### Focus groups

Focus group data were analysed using thematic analysis (NVivo11). Findings showed consistency in the way all HBF participants (clients, volunteers and staff) articulated their reflections on the impact of creative activities. The benefits of creative activities were highlighted in clusters emerging from around three main overarching themes: 'skills', 'social aspects/friendships' and 'mood-personal sphere'; the latter subdivided into 'brain' (creative thought processes), 'routine', self-expression' and 'confidence'. During focus groups, coresearchers were asked to write answers to the questions posed, and a collective discussion followed. Clients responded consistently about benefits of the creative programmes. One client listed 'Activities help my brain think about other stuff and keep me busy and relaxed. You can't think about bad things. To learn more skills. I can use skills to help others'.

Learning new or improving existing skills was a key theme emerging out of focus groups, as skill development appeared to change clients' perceptions of their status; as a female volunteer at HBF for 2 years suggested 'participants do not feel assessed as clients, they are just learners or artists. These groups paved the way for identifications different from being a victim, a refugee or an asylum seeker'. Clients also mentioned how acquiring new skills was an important factor in improving self-esteem; one client wrote about attending CAP 'Give me confidence. Make me feel good and useful. Change my mood. Learn new things and meet people. Improve my skills. Give me hope to lead a better life'.

Learning new skills was a benefit connected to the second theme that emerged during focus groups, the social aspect. It allowed people to meet others in similar situations and create friendships and/or a support network; as a client wrote 'Allows me to feel included. Safe environment. Improves my mood. Learn new skills. Gets me out of the house. Meet new people'. Staff too recognised the social aspect of activities as a major benefit, with one suggesting 'in these groups, clients learn to have balanced, reciprocal relationships, which they have not experienced before or for a long time given their experience of violence (trafficking). They provide clients with an alternative identity, that is, they are not simply victims here, they are learners, artists, dressmakers, etc.' Similar themes emerged in the volunteer focus group during which a relatively new male volunteer involved of 4 months was impressed by the solidarity between clients, emphasising the benefits of the 'feeling of belonging to

a group' and 'strong support that people receive from other members'.

The third and final theme that emerged when discussing the benefits of CAP the mood-personal sphere. Staff commented on the mood enhancing benefits of CAP for people with PTSD; one staff member observed 'creative activities allow clients to momentarily leave out their problems or memories, by focusing on a particular activity they cannot think about anything else'. The focus group with staff emphasised that being busy in the company of others allowed clients to feel safe and not to dwell on their own situation. In concentrating on learning new skills, clients blocked out (although for short periods) memories of the past. Importantly, responses illustrated that focus group participants perceived their involvement as beneficial at multiple and overlapping levels. Enhancement of mood appeared to derive from not only meeting new people, forging meaningful relationships and feeling part of a group but also connected to a growing self-confidence and self-esteem experienced as clients acquired new skills.

### Interviews

During interviews, clients elaborated on practical and social skills, and commented on the positive effect of attendance on their mood and emotion:

Practical skills: learning new practical skills or improving those previously learnt emerged strongly among asylum seekers without the right to work. As a client pointed out:

I have learned a lot of skills: I've learned sewing, now I'm learning art, I can do better, before I didn't know how to draw a person, to mix colours, you know such things… so you keep on learning, slowly by slowly, then in the end you find yourself, you're a professional, so that is very great… to me… I like to come, always and attend, and listen to what they tell me, and I do it.

Or, as another client explained:

I've learned new skills […] and you can never waste your time learning new skills. New skills always help you in your life, always. So, everything I've learned will help me. You think like 'film club, learning how to edit, etc.'. People pay people on YouTube to edit their films, and I'm, like, I could just do my own, if I wanted to have a YouTube channel, I can do my own.

Skill improvement provided a sense of achievement and improved self-confidence, talking about learning new skills, one client observed:

When you are in this situation it feels like life has, in a way, stopped. And you can't do anything to change it. But by doing all this activity you feel, or I felt … it was like, you know, that I was learning something in my life, rather than just waiting.[…] For instance, if I go for a job somewhere, where you have to write what skills you have, so I could include all of this. I mean I

don't have a certificate or, like, proper qualifications, but I've all those skills, so I feel like, it is, in a way, impressive? Because you feel like, you know, rather than just waiting and not doing anything, you have been learning.

The sense of self-improvement associated with learning skills was linked to the perception that by learning, clients were preparing themselves for a new life when they would be allowed to live and work in the UK, countering the perception that life for them had stopped and giving them something to look forward to. The process of skills learning was not simply a vertical one; peer learning was equally important, stressing the importance of social aspects.

Social skills: the opportunity to overcome social isolation and create a social network emerged as one of the benefits of creative engagement. As one client indicated:

> The groups changed me because you know when I come, I feel lonely and doesn't have anybody. I was lonely I don't even know whether there will be a home from morning to live in. But when I started coming these groups, I told you I feel active, I feel coming to see people, that I speak to. I don't even know how to communicate with people before. I told you because of my coming here I now feel to communicate to people. Before I never communicate to people, I don't know how to do it. And I don't know how to play with people. But these groups make me now taught me: play with people, to meet me.

The client expressed the difficulties of establishing new relationships and the feeling that the group was a place in which they could regain this ability to connect with other people. The social aspect was an important point reiterated in all interviews; fellow clients were described as 'friends' but also as 'brothers and sisters'. Arts activities allowed clients to create a community characterised by solidarity, as one client explained:

> If you're in a situation where you've been completely isolated from people for a while and you just don't know who to trust, or to be around people, it's one of those spaces where you can get to meet people, socialise, and actually make friends.[…] And these friendships do last. A life-time, because you understand what the person has gone through, you don't have to explain a lot, you know, they just know that you've been through something terrible.[…] So, it's easier when you have people here who just understand what you've possibly been through.

This remark connects with the observation that 'engagement in participative creative arts activities in communities can help to build social capital, address loneliness and social isolation, and build personal confidence and a sense of empowerment' (Staricoff, p32).[25] The awareness of experiencing similar situations makes clients feel free

to share their own experiences with one another but also not to, if they did not feel like opening up:

> There is a sense of community: you know you go there, you know there are people like you in the same situation you're not going to be judged, so it's that sense of community: we know what's going on, we don't have to talk about it.[…] It's a distraction from immigration and we know we are all going through it, but we don't have to talk about it. There are other things going on in life, and we talked about, for example in the art group, the works we produced and what we could do, what we could achieve and get inspired by each other's work.

Again, this remark introduced the third broad theme which emerged out of the interview analysis, which was the impact of creative arts on personal mood and emotion.

Mood and emotion: learning or improving skills while forging long-lasting relationships in a context of deep social isolation (outside of HBF), inevitably had a substantial impact on mood and emotion. In discussing this aspect of creative arts, interviewees focused particularly on how groups allowed them to have a routine, a 'luxury' one client said 'when you can't do nothing but waiting for the government to decide on your right to stay in the country'. Routine combined with skills learning/improving was thought to have a positive impact on clients' mental health; 'having something to do' and 'something to look forward to' were recurrent expressions during interviews. As one client noted:

> I used to come here every day of my life: Monday, Tuesday, Wednesday… because I don't have anywhere to go, so I just come here, and I attached myself to the groups, we met, we talk, we… then I've gone to knit, I've gone to do computer, reading, many things I learnt here, so I just like it.

Another client elaborated further:

> It's something that I look forward to. One thing you have to bear in mind is that I am not working, and I am not studying, you know, so it was the only thing that I would look forward to because it was something to do, otherwise I would just be at home, doing nothing, you know, feeling very sorry for myself, getting upset all the time. I mean I still feel that way but at least there's something to look forward to, when you don't have hope or something to look forward to, it increases your depression levels, so for me it's helped me greatly.

This statement describes the social and emotional isolation as part of postmigration living difficulties. The routine of classes, together with an awareness that others in similar situations would attend, provided comfort to clients, countering feelings of isolation:

> There is nothing else for you and it's also a way of, like, using your time wisely, 'cause at the end of the

day most of the time you're just sitting down doing nothing, and doing nothing slowly begins to affect your mind, your brain, you know you become lazy, you know you literally are just in a four-squared room[…]

The benefits of a routine were increased by inclusion of creative activities that according to clients and volunteers, had a positive impact on mood because they exercised the brain and allowed clients to find a new language to express themselves. As a client stated:

[Attending creative groups] gives you a lot of confidence, you know. And it allows your brain to think outside of the box because I think we are just in that box and… but when you attend these classes, your mind, you know, opened and you learn new things and you want to then learn other things as well.

This comment highlights an important link between arts and feelings; as another stated:

It is more of expressing, more of letting go, it's like getting a spirit out of you, and you don't have necessarily have to tell someone 'I've done this because of that and that, and this', you know. Because sometimes you just don't find the voice to talk about it and, I, as a person am really shy and, you know, I feel easily embarrassed, you know.[…] So… that's why I'm into arts, yes I'm doing arts really.

Clients agreed that the possibility of self-expressing without having to articulate their feelings, helped them grow emotionally and gain confidence. It was not only clients who benefited from CAP; findings also pointed to the positive impact on volunteers, a point stressed by all volunteers when discussing personal experiences.

## DISCUSSION

Creative and cultural activities were observed over a sustained period of 5 months. Clients participating in the activities had reached the integration stage of the model of integrated care consequently had been taking part in CAP activities for 2 plus years prior to the research. Activities provided refugees and asylum seekers with new skills, including practical and technical skills, and social and life skills involving language acquisition obtained partly through informal peer learning and mutual support. Learning new skills contributed to the sense of well-being and empowerment experienced by HBF clients. The time spent by the researcher at the HBF was vital for familiarisation with the context, getting to know people involved in the organisation and explaining the research to recruit participants from among clients, volunteers and staff. The fact that several clients attended more than one group per week (2–4) made it possible to forge trusting relationships and recruit coresearchers, allowing them to feel comfortable in speaking their mind. A possible issue here was that the number of groups attended by clients was not accounted for, so it was not evident as to whether increased participation furthered benefits at a higher/faster level. Once recruited, participants remained with the study in their coresearcher roles and continued to attend activities on a weekly basis for the 5 months (except for rare absences due to illness).

Activities appeared to positively enhance mood and emotion both for clients and volunteers facilitating CAP groups. Although passive participation was not compared,[28] active and creative participation in the groups specifically benefited the clients. A key reason was that asylum seeker status in the UK would not have permitted alternative occupation or employment, so clients might not have otherwise left their homes or met people on a regular basis. In keeping with previous research, participants reported that participation in cultural and social activities contributed to their social health,[18] and aligned with a growing body of evidence[1] to suggest that creative activities enhanced mental well-being,[19–27] helping clients to develop self-confidence and resilience.[21] Client preferences for activities indicated that singing was their favourite group, followed by photography, art and textiles. Singing was regarded as beneficial because it allowed people to meet socially and work collaboratively, aspects considered important in countering loneliness. Findings aligned with a recent qualitative study indicating that choral singing promoted improvements in social, emotional, physical and cognitive functioning and that benefits were experienced similarly irrespective of age, gender and nationality.[40] In the current study, photography, art and textiles were praised by clients for the skills taught and although social aspects were relevant, skills that allowed participants to continue activities at home (eg, sewing and drawing) were highly valued. Interviewed staff were asked to comment on the popularity of activities, and although the order was the same as that of clients, they were not asked their opinion on which were most beneficial.

The current study bridged the gap between two research disciplines: arts-in-health and forced displacement and mental health. It contributed to existing literature by demonstrating beneficial effects of creative activities on the well-being of refugees/asylum seekers and showed that PAR was an appropriate and 'democratic' means of collaboration between displaced communities and academics (Bradbury, p3).[39] In contrast with traditional research where 'academics benefit from the research, but often the people involved hear very little if anything from the researcher again' (Daykin and Stickley, p78),[33] a core element of PAR is that research should be meaningful and have tangible outcomes for coresearchers as they work together to bring about significant change within their community or society at large. In the current study, client involvement in the research and their decision to organise a public exhibition to showcase their artwork was a means for them to interact with wider society and actively rewrite the narrative around asylum seekers and refugees, often depicted in negative terms in the public sphere.[41]

Adopting a PAR approach is important when working with immigrant communities as it ensures that the research question is of 'genuine importance' (Vaughn and Jacquez, p118),[34] and facilitates translation of research findings into 'actions that will benefit the community' (Vaughn and Jacquez, p119).[34] PAR was particularly useful when conducting research with refugees/asylum seekers with PTSD and other mental health issues because the process of involvement as coresearchers countered client isolation and built social capital through regular meetings. In helping to interpret findings, clients acquired research methods skills (eg, drafting interview guides, conducting focus groups and discussing data) and organisational abilities (event/exhibition). Participating in the research and disseminating findings gave clients a sense of agency and ownership of the project that enhanced self-confidence and feelings of empowerment, one of the pillars of PAR especially when community based (Blumenthal, p2).[35] Participation provided a positive means of expression countering their refugee/asylum seeker status that otherwise, rendered them effectively powerless.

In terms of study weaknesses, the greater number of female than male participants was probably due to the fact that most creative groups were led by female volunteers (2 m, 8f). Most clients attending these groups were also female except for singing, where numbers were approximately equal (although only one male choir member took part in a focus group). Given the nature of the singing activity, conversation was limited to the first minutes before arrival; once rehearsals had started, it was difficult for the researcher to form relationships with clients. It was also possible that some activities were seen by clients as gender-neutral (eg, photography or singing), but others such as working with fabrics and soft materials (eg, wool and cloth) may have been seen as a predominantly female activity; male participants might have been more comfortable working with harder materials (eg, wood and metal). It was generally more difficult to engage male participants in the research; a possible reason for this was female clients might have found it easier to relate to the female researcher. A further weakness was low attendance for focus groups possibly due to difficulties of organising them at convenient times for everyone, given changing client circumstances (eg, interviews for benefits and housing). More clients expressed an interest than those who participated, nevertheless were happy to informally discuss topics debated. Others wanted to be kept in the loop without participating in focus groups and interviews, so research methods were discussed informally during meetings between researcher and clients.

## CONCLUSIONS

This study expanded on arts-in-health and collaborative research approaches by exploring the effects of cultural and creative activities on the health and well-being of refugees and asylum seekers. Creative activities at HBF provided opportunities for clients to socialise and develop new relationships based on respect and mutual recognition. In turn, these opportunities countered loneliness and allowed refugees/asylum seekers to develop a sense of belonging in a safe space. The opportunities further supported freedom of expression and the possibility of developing a new identity to counter the negative stereotype of being viewed as a refugee or asylum seeker,[41] leading to sustained improvements in positive mood and self-confidence, and countering anxiety, depression and other mental health issues. By focusing on the relationship between arts, well-being and forced displacement and in keeping with PAR, the study actively changed the narrative surrounding refugees/asylum seekers and gave an understanding of how to improve conditions for forcibly displaced people through opportunities for social connectivity, creative thinking and employment skills. It is recommended that other refugee/asylum seeker organisations could use the interview questions generated by the study or develop their own, following PAR guidelines, to appraise and improve migrant provision particularly in terms of upskilling peer-learning. If scaled up across the UK, programmes of creative activities for migrants could reduce pressure on health and social care services and help to address the social determinants of health inequality.[16] Research outputs have wider implications for policy development, for example by the United Nations Refugee Agency, WHO and other government and non-government agencies.

**Twitter** @h_chatterjee

**Contributors** CC, LJMT and HJC made substantial contributions to the conception and design of the work; drafting and critically revising the work for intellectual content and the final approval of the version published. CC conducted the acquisition, analysis and interpretation of data for the work. All authors agree to be accountable for all aspects of the work. We would like to thank the clients, volunteers and staff who were coresearchers in the project.

**Funding** The work was supported by the Global Challenges Research Fund (GCRF) Grant reference: ES/P003818/1 PI: HJC.

**Competing interests** None declared.

**Patient consent for publication** Not required.

**Ethics approval** The study gained approval from UCL Ethics Committee (Ethics Application 4526/002 Codeveloping a method for assessing the psychosocial impact of cultural interventions with displaced people: towards an integrated care framework) to carry out the research with potentially vulnerable participants.

**Provenance and peer review** Not commissioned; externally peer reviewed.

**Data sharing statement** Due to the confidential nature of the human participant data, it will not be freely available via free access databases. Any request for data should be addressed directly to the corresponding author.

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
