## [Reviewer comments · BMJ Open]

BMJ Open

BMJ Open is committed to open peer review. As part of this commitment we make the peer review history of every article we publish publicly available.

When an article is published we post the peer reviewers' comments and the authors' responses online. We also post the versions of the paper that were used during peer review. These are the versions that the peer review comments apply to.

The versions of the paper that follow are the versions that were submitted during the peer review process. They are not the versions of record or the final published versions. They should not be cited or distributed as the published version of this manuscript.

BMJ Open is an open access journal and the full, final, typeset and author-corrected version of record of the manuscript is available on our site with no access controls, subscription charges or pay-per-view fees (<http://bmjopen.bmj.com>).

If you have any questions on BMJ Open's open peer review process please email info.bmjopen@bmj.com

BMJ Open

Assessing the impact of artistic and cultural activities on the health and wellbeing of forcibly displaced people using participatory action research

Journal:	BMJ Open
Manuscript ID	bmjopen-2018-025465
Article Type:	Research
Date Submitted by the Author:	18-Jul-2018
Complete List of Authors:	Thomson, Linda; University College London Division of Biosciences Clini, Clelia; Loughborough University - London, Institute for Media and Creative Industries Chatterjee, Helen; University College London, Division of Biosciences
Keywords:	creative activities, forced displacement, participatory action research, post-traumatic stress disorder, refugees and asylum seekers

Peer Review Only

1 Assessing the impact of artistic and cultural activities on the health and wellbeing of forcibly
displaced people using participatory action research

Clelia Clini, Linda J Thomson, Helen J Chatterjee

Correspondence to: Helen J Chatterjee, UCL Division of Biosciences, 507B Darwin
Building, University College London, London, WC1E 6BT, UK. email
8 h.chatterjee@ucl.ac.uk telephone +44(0)2031084104 fax +44(0)2076797193

Clelia Clini, Institute for Media and Creative Industries, Loughborough University London,
UK

Linda J Thomson, UCL Division of Biosciences, University College London, London, UK

Key words: creative activities; forced displacement; participatory action research; post-
traumatic stress disorder; refugees and asylum seekers;

Word count: 4131

**Abstract**

Objective: Drawing upon a growing body of research suggesting that taking part in artistic
and cultural activities benefits human health and wellbeing,[1-8] the objective was to develop
a participatory action research (PAR) method for assessing the impact of arts interventions on
forcibly displaced people. Although creative activities are seen to provide major benefits for
refugees and asylum seekers, researchers have become ‘increasingly conscious of the value
of recording and analysing what has been happening’.[9 p1] Many European organisations
provide creative activities aiming to reduce social isolation, as displaced people often display
depression, anxiety, post-traumatic stress disorder and non-affective psychosis.[10-12]

Design: A study was conducted following PAR principles comprising observation, focus
groups and in-depth semi-structured interviews.

Setting. London-based charity working with asylum seekers and refugees.

Participants: Refugees/asylum seekers (n=7), volunteers (n=7) and charity staff (n=8).

Results: Qualitative data was analysed using NVivo 11. Focus group findings showed that
participants articulated reflections on the impact of creative activities around three main

themes: skills, social engagement, and personal emotions which were explored during
interviews. Findings showed that artistic and cultural activities impacted positively on
participants by helping them to find a voice, create a support network, and learn practical
skills useful in the labour market.

Conclusions: The study expanded on arts and wellbeing research by exploring effects of
cultural and creative activities on the psychosocial wellbeing of refugees and asylum seekers.
By focusing on the relationship between arts, wellbeing and forced displacement, the study
was instrumental in actively trying to change the narrative surrounding refugees and asylum
seekers,[45] often depicted in negative terms in the public sphere.

Strengths and limitations of this study:

- • Focus on collaborative research
- • Different cultural and artistic activities studied
- • Researcher spent five months in research setting
- • Approximately twice as many females participated
- • Focus groups attracted relatively low numbers

18 **Introduction:**

The recent All-Party Parliamentary Group on Arts Health and Wellbeing Inquiry Report
shows a growing acknowledgement of the impact of arts and creative practices, stating ‘arts
engagement has a beneficial effect upon health and wellbeing and therefore has a vital part to
play in the public health arena’.[13 p11] More than 70 years ago, the World Health
Organisation defined health as a ‘state of complete physical, mental and social wellbeing and
not merely the absence of disease or infirmity’.[14 p2] As a key element of health, the notion
of achieving increased wellbeing is central to many studies aimed at improving quality of
life.[15 p5] Currently, consensus among practitioners defines wellbeing as ‘the dynamic
process that gives people a sense of how their lives are going through the interaction between
their circumstances, activities and psychological resources or mental capital’.[16 p3] Rather
than depending on a single factor, wellbeing is seen as the result of interplay between
different elements, distinguishing personal wellbeing (emotional wellbeing, satisfying life,
vitality, resilience, self-esteem and positive functioning) from social wellbeing (supporting
relationships, trust and belonging).[17] Other authors emphasise ‘resilience and flourishing,
rather than just surviving’.[18 p243] a crucial point when investigating the situation of

1 forcibly displaced people who often survive in extreme conditions and whose experience is
2 often discussed in terms of dehumanisation.[19,20]

Research conducted in the field of forced displacement and mental health shows that
‘asylum seekers and displaced people report high rates of pre-migration trauma’,[20 p275]
and significant levels of post-traumatic stress disorder (PTSD), depression and anxiety.[10,
12, 21-23] In addition to pre-migration trauma, refugees and asylum seekers often encounter
post-migration living difficulties, such as socio-economic disadvantage, employment barriers,
social and emotional isolation, racism and hostility, experience of detention, and uncertainty
related to the asylum application process.[10, 12, 21, 24] If, as authors suggest, there is a
strong link between social and economic conditions in terms of income, employability,
housing and mental health,[25-29] then asylum seekers and refugees are at greater risk of
developing mental health issues than the general population.

Arts-and-health practitioners believe that ‘aesthetics act upon our senses to make us feel
more, hear more and see more than we otherwise might... feelings are intertwined with
mental, physical, spiritual and social health’.[30 p4] Several studies explored the relationship
between arts, health and wellbeing and a growing body of evidence[13] suggests that cultural
participation (including music, art making, theatre, dance, museum and heritage activities)
enhances human health and wellbeing.[1-8, 31] Research conducted in museums-and-health,
for example, suggests that ‘museum and art gallery encounters can help with a range of health
issues, enhance wellbeing, and build social capital and resilience’[3 p286]. According to
research conducted in Norway involving over 50,000 adults,[32] participation in both
receptive/passive and creative/active cultural activities was significantly associated with good
health, life satisfaction, and low anxiety and depression. Findings demonstrated how creative
practices, such as knitting, had a positive impact on people suffering from depression and
post-traumatic stress disorder because ‘the movements involved in knitting are bilateral,
rhythmic, repetitive, and automatic’.[32 p40] The authors attributed the positive mood of
knitters to enhanced production of serotonin resulting from repetitive movements and that
bilateral processes appeared to engage brain capacity and facilitate a meditative-like state
more readily than unilateral ones.

The current study was conducted in the light of mental health and arts/museums-and-
health evidence given the large number of organisations that engage migrants, refugees and
asylum seekers in the arts, with nearly 200 of these in the UK.[9] The current research was
conducted at one such organisation, Helen Bamber Foundation (HBF), a charity offering
support to refugees and asylum seekers as victims of torture and human rights violations.

HBF adopts a holistic approach to support its clients that includes cultural and creative
activities within a model of integrated care. The model is based on the needs of traumatised
asylum seekers and refugees who often require simultaneous assistance on a variety of fronts
(e.g. psychological support might not be effective if a person does not have a place to sleep or
lives in a dangerous situation, though giving people a place to sleep is not enough to
overcome trauma). Their model encompasses services such as legal and health support
(psychological and physical), welfare and housing advice, and a creative arts programme
(CAP) including art, photography, knitting and textiles, singing, English and computing, led
by professional volunteers and attended by nearly 100 clients. By focusing on the relationship
between arts, wellbeing and forced displacement, the study aimed to determine why
participation in creative arts enhanced the sense of wellbeing experienced by HBF clients,
and contribute to the current debate on migration and public health.

**Method**

**Design**

Qualitative research conducted comprised participant observation, focus groups and one-to-
one semi-structured interviews.[33] The research was operationalised according to the
principles of participatory action research (PAR) as a ‘way of opening up space for dialogue
and conversation’,[34 p28] aiming ‘understand and also improve a particular situation’,[35
p1] The approach encourages an active contribution in the production of knowledge within a
collaborative framework, emphasising ‘equal partnerships’,[36 p167] and the ‘role of the
participant in the design, implementation, and dissemination of the research’.[37 p78] The
collaborative character of PAR is grounded in efforts to ‘democratise the research
process’,[35 p3] so it is valuable for everyone involved and where ‘equal weight and
consideration is given to the contributions of both the community and academic partners’[36
p170]. By offering equal weight to contributions, participants develop a sense of ownership,
which in turns fuels the feeling of empowerment, one of the pillars of PAR, especially when
community-based.[39]

**Participants**

The research recruited an opportunity sample of participants who, in keeping with PAR were
called ‘co-researchers’ and who had differing involvement in the project depending on their
role at HBF (clients, volunteers or staff). Information leaflets and researcher contact details

were placed in reception to encourage participation and clients were recruited directly by the
researcher while based at HBF. The criteria for recruitment were that clients had received
therapy for at least two years prior to the research and, while still vulnerable, had entered the
integration stage of the model of integrated care, and that they attended at least one of the five
CAP groups (art, knitting, photography, singing and textiles). Volunteers and staff were
recruited via email and personal contact. Co-researchers (n=22: 7m) participated in focus
groups comprising clients (n=6), volunteers (n=4) and staff (n=8), and interviews (n=18)
comprising clients (n=7), volunteers (n=7) and staff (n=4).

10 Patient and public involvement

The development of the research questions and outcome measures about the effects of cultural
and creative activities on the psychosocial wellbeing of refugees/asylum seekers was
informed by HFB clients, volunteers and staff taking part in PAR. Focus groups and
interviews gave voice to participants' priorities, experience and preferences but, due to the
vulnerability of clients, no personal information was requested as the process of remembering
could have been difficult for PTSD sufferers, [10, 22] and although some participants
mentioned personal issues and experiences, they were not encouraged to do so. Participants
were involved as co-researchers within the research collaboration and data generated by their
involvement informed focus groups and interviews. Participants were indirectly involved in
the recruitment process by recommending that other people join them, though the researcher
needed to ensure they matched recruitment criteria (above). Participants were directly
involved in conducting the study and determining the interview questions. Results were
disseminated to participants through an end-of-project event and on-going communication
with the researcher, and an internal report will be lodged at HBF.

26 Data Collection

Data collection carried out by the location-based researcher consisted of three stages: i)
participant observation (months 1-2), ii) focus groups (months 3-4) and iii) semi-structured
interviews (month 5). Participant observation was used as a starting point 'for studying how
organisations work, the roles played by different staff and the interaction between staff and
clients'. [33 p32] In stage i) the researcher attended creative classes for clients and spent a day
a week working from HBF (12-15 hours per week) to determine the nature of focus groups.
In stage ii) four focus groups were held, the first and final with clients from arts,
photography, singing, and textiles, the second with volunteers, and the third with staff. In the

focus groups, participants were asked to discuss research questions (Table 1) involving ways
of gathering information on the impact of creative activities, and reasons for attending groups
(or in the case of volunteers and staff, reasons for involvement). All focus groups were
organised informally to facilitate development of discussion among participants and allow an
exchange of experiences and ideas. Ideas which emerged during focus groups laid the basis
for the first drafts of interview questions, which were tailored to clients, volunteers or staff
(Table 2). In line with the collaborative ethos, drafts were circulated to receive feedback from
co-researchers prior to the interview stage, and volunteers and staff were consulted and asked
to provide comments and suggestions on the questions. For stage iii) interviews carried out
with were semi-structured to allow free expression within the research constraints and limit
potential influence of the researcher.[33,34]

INSERT TABLE 1 ABOUT HERE

INSERT TABLE 2 ABOUT HERE

17 Data analysis

A first informal thematic analysis was conducted by the HBF-based researcher while writing
field notes consisting of a chronicle of descriptive rather than analytic events to provide the
'raw material of the research'. [33 p38] The process of transcribing notes was used to identify
key points connected to the research question. This preliminary analysis informed the focus
group topics where themes were discussed more in detail. After the first round of focus
groups, all notes were compiled into a Word document and repeatedly read in search of
recurring subjects and themes. Thematic analysis of focus group outcomes were explored in
detail during interviews. All interviews were recorded and transcribed to become familiar
with the data gathered and begin the coding process [40]. In the transcript, all details were
recorded verbatim (e.g. sighs, laughter, silences, and tears) as their exclusion could have
changed the meaning expressed.[41] Transcriptions were uploaded into NVivo11 to produce
nodes from coding-relevant concepts. NVivo's word frequency query facilitated the search
for key words (including stem words and synonyms) grouped together under the same theme.
The text search query provided a comprehensive analysis of data. After coding of data into
initial themes,[40] a review of themes followed to refine the analysis.

Results

Focus Groups

Thematic analysis of focus group data showed consistency in the way all HBF participants
(clients, volunteers and staff) articulated their reflections on the impact of creative activities.
In discussing the value of engaging in creative arts, participants highlighted the benefits of
activities as:

- • Learning new practical skills or improving those previously learnt;
- • Acquisition of technical, language, social and life skills;
- • Development of new social relationships based on respect and mutual recognition;
- • Opportunities to meet people, forge friendships, counter loneliness and develop a sense of
belonging in a safe space;
- • Value of peer-learning and mutual support;
- • Improvement in mood and self-confidence; and
- • Freedom of expression and developing a new identity other than refugee/asylum seeker.

Thematic clusters emerged from around three main overarching (and overlapping) themes
from these benefits: 'social aspects-friendships', 'skills', and 'mood-personal sphere'; the
latter was sub-divided into 'brain' (how creative activities helped the brain), 'routine', self-
expression' and 'confidence'.

Interviews

During the interviews, respondents tended elaborated on skills, both practical and social, and
also commented on the effect that attendance had on their mood and emotion.

Learning new practical skills: One of the themes which emerged strongly during interviews
was that creative arts groups allowed participants to learn new skills and/or to improve prior
skills. Skill improvement was important for asylum seekers without the right to work. As a
client pointed out:

*I have learned a lot of skills: I've learned sewing, now I'm learning art, I can do*
*better, before I didn't know how to draw a person, to mix colours, you know such*
*things... so you keep on learning, slowly by slowly, then in the end you find yourself,*

*you're a professional, so that is very great... to me... I like to come, always and attend,*
*and listen to what they tell me, and I do it.*

Or, as another client explained:

*I've learned new skills [...] and you can never waste your time learning new skills. New*
*skills always help you in your life, always. So, everything I've learned will help me. You*
*think like "film club, learning how to edit, etc.". People pay people on YouTube to edit*
*their films, and I'm, like, I could just do my own, if I wanted to have a YouTube channel,*
*I can do my own.*

The improvement of skills was a benefit that provided a sense of achievement and improved
self-confidence, illustrating the overlapping nature of themes that emerged. Talking about
new skills learnt, one client observed:

*When you are in this situation it feels like life has, in a way, stopped. And you can't do*
*anything to change it. But by doing all this activity you feel, or I felt...it was like, you*
*know, that I was learning something in my life, rather than just waiting. [...] For*
*instance, if I go for a job somewhere, where you have to write what skills you have, so I*
*could include all of this. I mean I don't have a certificate or, like, proper qualifications,*
*but I've learned all those skills, so I feel like, it is, in a way, impressive? Because you*
*feel like, you know, rather than just waiting and not doing anything, you have been*
*learning.*

Self-improvement was seen not only as a result of learning from professionals, but also from
peer-learning; several clients commented on the fact that they learnt from friends as well as
volunteers, pointing to the importance of social aspects of the classes.

Acquisition of technical, language, social and life skills: One of the benefits of being engaged

[revised manuscript text omitted]

among clients, volunteers and staff for focus groups and interviews. Moreover, the fact that
several clients often attended more than one group a week (in certain cases all five groups)
made it possible to forge trusting relationships and recruit co-researchers, allowing them to
feel comfortable in speaking their mind regarding the research.

In keeping with previous research, creative and cultural participation lead to improved
social health [30] and aligned with a growing body of evidence[13] to suggest that the five
selected CAP activities enhanced mental wellbeing[1-8, 31] helping clients to develop self-
confidence and resilience.[3] A possible issue here was that the number of groups attended by
clients, that varied from one to three throughout the week, was not taken into account, so it is
not evident as to whether increased participation would have enhanced positive aspects of
wellbeing at a higher or faster level. Although passive participation was not compared,[32]
active and creative participation in the CAP groups specifically benefited refugees and
asylum seekers, as their status in the UK would not have permitted alternative occupation or
employment, so may not have left their homes and met other people on a regular basis.

In terms of study weaknesses, the greater number of female participants (approximately
twice that of males) was due to the fact that most creative arts groups were led by female
volunteers (only two of ten volunteers were male) and most participants in these groups were
also female except for the singing group, where numbers were roughly equal. Consequently,
it was easier for the female researcher to forge relationships with women rather than with
men. With the singing group, while one male choir member took part in one focus group, it
was difficult to engage male participants in the research; given the nature of the activity,
conversation was limited to the first minutes before arrival as once rehearsals started, it
became difficult to forge meaningful relationships with participants.

A further weakness was the low level of attendance for focus groups. As a result of
undergoing difficult situations, clients in particular had circumstances that could change

quickly, consequently, not all of those who expressed an interest in participating were able to
do so but most were nevertheless happy to informally discuss the topics debated afterwards;
other clients expressed an interest in being kept in the loop regarding the progress of the
research. As for volunteers, while at least one volunteer per group was involved in the
research at any point, it was difficult to organise focus group times convenient for everyone,
so again themes and research methods were discussed in more informal ways with the
researcher based at HBF during weekly meetings with clients.

9 **Conclusions**

This study expanded on arts and wellbeing research by exploring the effects of cultural and
creative activities on the psychosocial wellbeing of refugees and asylum seekers. The
activities at HBF provided opportunities for clients to meet people and develop new social
relationships based on respect and mutual recognition. In turn, these opportunities countered
loneliness and allowed refugees and asylum seekers to develop a sense of belonging in a safe
space. The opportunities further supported freedom of expression and the possibility of
developing a new identity to counter the negative stereotype of being viewed as a refugee or
asylum seeker, leading to sustainable improvements in positive mood and self-confidence,
countering anxiety and depression, and other mental health issues. By focusing on the
relationship between arts, wellbeing and forced displacement and in keeping with PAR, the
study has been instrumental in actively trying to change the narrative surrounding refugees
and asylum seekers,[42] often depicted in negative terms in the public sphere. Refugee and
asylum seeker organisations should consider commissioning PAR to involve their clients,
volunteers and staff in appraising and improving the current provision for migrants to the
UK, particularly in terms of broad-ranging skill sets and psychosocial support.

26 27 **Contributors**

CC, LT and HC made substantial contributions to the conception and design of the work;
drafting and critically revising the work for intellectual content; and the final approval of the
version published. CC conducted the acquisition, analysis and interpretation of data for the
work. All authors agree to be accountable for all aspects of the work. We would like to thank
the clients, volunteers and staff who were co-researchers in the project.

Funding

The work was supported by the Global Challenges Research Fund (GCRF) Grant reference:
ES/P003818/1 PI: HC

Ethics

The study gained approval from UCL Ethics Committee (Ethics Application 4526/002 Co-
developing a method for assessing the psychosocial impact of cultural interventions with
displaced people: Towards an integrated care framework) to carry out the research with
potentially vulnerable participants.

Competing interests

The authors have no competing interests.

Data sharing statement

Due to the confidential nature of the human participant data, it will not be freely available via
free access databases. Any request for data should be addressed to the corresponding author.

References

- 1. Bygren LO, Johansson S, Konlaan BB, Grijbovski AM, Wilkinson AV, Sjöström M.
Attending cultural events and cancer mortality: A Swedish cohort study. *Arts Health: Int J*
*Res Pol Pract* 2009;1:64–73. doi:10.1080/17533010802528058
- 2. Camic P, Chatterjee H. Museums and art galleries as partners for public health
interventions. *Perspect Public Health* 2013;133:66–77.
- 3. Chatterjee H. Museums and art galleries as settings for public health interventions. In S
Clift, P Camic (Eds.), *Oxford Textbook of Creative Arts, Health, and Wellbeing:*
*International Perspectives on Practice, Policy and Research*. New York, Oxford: Oxford
University Press 2016; p.281–289.
- 4. Clift S, Skingley A, Coulton S, Rodriguez J. The effectiveness and cost-effectiveness of a
participative community singing programme as a health promotion initiative for older
people: protocol for a randomised controlled trial. *BMC Public Health* 2011;11:1–6.
doi:10.1186/1471-2458-11-142
- 5. Clift S, Camic P. *Oxford Textbook of Creative Arts, Health, and Wellbeing: International*
*Perspectives on Practice, Policy and Research*. New York, Oxford: Oxford University
Press, 2016.

- 6. Konlaan BB, Bygren LO, Johansson S. Visiting the cinema, concerts, museums or art
exhibitions as determinant of survival: a Swedish fourteen-year cohort follow-up. *Scand J*
*Public Health* 2000;28:174–178. doi:10.1177/14034948000280030501
- 7. Staricoff RL. *Arts in health: A review of the medical literature*. London: Arts Council
England, Research Report 2004;36:1–91. [http://www.artsandhealth.ie/wp-](http://www.artsandhealth.ie/wp-content/uploads/2011/08/AHReview-of-Medical-Literature1.pdf)
[content/uploads/2011/08/AHReview-of-Medical-Literature1.pdf](http://www.artsandhealth.ie/wp-content/uploads/2011/08/AHReview-of-Medical-Literature1.pdf)
- 8. Staricoff RL. *Arts in health: The value of evaluation*. *Perspect Public Health*
2006;126:116–120. <http://journals.sagepub.com/doi/pdf/10.1177/1466424006064300>
- 9. Kidd B, Zahir S, Khan S. *Arts and Refugees: History, Impact and Future*. London: Arts
Council England, 2008. [https://baringfoundation.org.uk/wp-](https://baringfoundation.org.uk/wp-content/uploads/2014/10/ArtsandRefugees.pdf)
[content/uploads/2014/10/ArtsandRefugees.pdf](https://baringfoundation.org.uk/wp-content/uploads/2014/10/ArtsandRefugees.pdf)
- 10. Katona C. Non-affective psychosis in refugees. *Br Med J* 2016;352:i1279.
doi:10.1136/bmj.i1279
- 11. Hollander A, Dal H, Lewis G, Magnusson c, Kirkbride JB, Dale H, Dalman C. Refugee
migration and risk of schizophrenia and other non-affective psychoses: cohort study of 1.3
million people in Sweden. *Br Med J* 2016;352:i1030. <https://doi.org/10.1136/bmj.i1030>
- 12. Robjant K, Robbins I, Senior V. Psychological distress amongst immigration detainees: A
cross-sectional questionnaire study. *Brit J Clin Psych* 2009;48:275–286.
doi:10.1348/014466508X397007
- 13. All-Party Parliamentary Group on Arts, Health and Wellbeing (APPGAHW). *Creative*
*Health: The Arts for Health and Wellbeing*. Inquiry Report. 2017;
<http://www.artshealthandwellbeing.org.uk/appg-inquiry/>
- 14. World Health Organization. *The Preamble of the Constitution of the World Health*
*Organization*. Geneva: World Health Organization, 1946.
[https://www.scielosp.org/scielo.php?pid=S0042-](https://www.scielosp.org/scielo.php?pid=S0042-96862002001200014&script=sci_arttext&tlng=pt)
[96862002001200014&script=sci_arttext&tlng=pt](https://www.scielosp.org/scielo.php?pid=S0042-96862002001200014&script=sci_arttext&tlng=pt)
- 15. White S, Blackmore C. *Cultures of Wellbeing: Method, Place, Policy*. Basingstoke:
Palgrave Macmillan, 2015.
- 16. New Economics Foundation (NEF). 2009. *National accounts of wellbeing: What is*
*wellbeing?* NEF. [http://www.nationalaccountsofwellbeing.org/learn/what-is-well-](http://www.nationalaccountsofwellbeing.org/learn/what-is-well-being.html)
[being.html](http://www.nationalaccountsofwellbeing.org/learn/what-is-well-being.html)
- 17. New Economic Foundation (NEF). *National Accounts of Well-being: Bringing Real*
*Wealth onto the Balance Sheet*. 2009.
https://neweconomics.org/uploads/files/2027fb05fed1554aea_uim6vd4c5.pdf

18. Ander E, Thomson LJ, Noble G, Lanceley A, Menon U, Chatterjee H. Generic well-being
outcomes: towards a conceptual framework for well-being outcomes in museums.
Museum Management Curatorship 2011; 26:237–259.
doi:10.1080/09647775.2011.585798
19. Crawley H, Düvell F, Jones K, MCMahon s, Sigona, N. Destination Europe?
Understanding the dynamics and drivers of Mediterranean migration in 2015. MEDMIG
Final Report 2016. www.medmig.info/research-brief-destination-europe.pdf
20. Vaughan-Williams N. “We are *not* animals!” Humanitarian border security and
9 zoopolitical spaces in Europe. Political Geography 2015;45:1–10.
doi:10.1016/j.polgeo.2014.09.009
21. Alpak G, Unal A, Bulbul F, Sagaltici E, Bez Y, Altindag A et al. Post-traumatic stress
disorder among Syrian refugees in Turkey: A cross-sectional study. Int J Psych Clinic
Practice 2015;19:45–50. doi:10.3109/13651501.2014.961930
22
22. Schubert CC, Punamäki R. Mental health among torture survivors: cultural background,
15 refugee status and gender. Nordic J Psych 2011;65:175–182.
16 doi:10.3109/08039488.2010.514943
23. Silove D, Sinnerbrink I, Field A, Manicavasagar V, Steel Z. Anxiety, depression and
18 PTSD in asylum-seekers: associations with pre-migration trauma and post-migration
stressors. Brit J Psych 1997;170:351–357. doi:10.1192/bjp.170.4.351
24. Aragona M, Pucci D, Mazzetti M, Geraci S. Post-migration living difficulties as a
21 significant risk factor for PTSD in immigrants: a primary care study. Italian J Public
Health 2012;9:e7525;1–8. doi:10.2427/7525
25. Allen J, Allen M. The social determinants of health, empowerment and participation. In S
Clift, P Camic (Eds.), Oxford Textbook of Creative Arts, Health, and Wellbeing:
International Perspectives on Practice, Policy and Research. New York, Oxford: Oxford
University Press; 2016, p27–34.
26. Fisher M. Why mental health is a political issue. The Guardian, July 16, 2012.
<https://www.theguardian.com/commentisfree/2012/jul/16/mental-health-political-issue>
27. Griffin J. The Lonely Society. London: The Mental Health Foundation.
https://www.mentalhealth.org.uk/sites/default/files/the_lonely_society_report.pdf
28. Marmot M, Allen J, Goldblatt P, Boyce T, McNeish D, Grady M et al. Fair Society,
Healthy Lives: The Marmot Review. 2010;
[http://www.instituteofhealthequity.org/resources-reports/fair-society-healthy-lives-the-](http://www.instituteofhealthequity.org/resources-reports/fair-society-healthy-lives-the-marmot-review/fair-society-healthy-lives-full-report-pdf)
[marmot-review/fair-society-healthy-lives-full-report-pdf](http://www.instituteofhealthequity.org/resources-reports/fair-society-healthy-lives-the-marmot-review/fair-society-healthy-lives-full-report-pdf)

29. Paul K, Moser K. Unemployment impairs mental health: Meta-analyses. *J Vocational*
Behavior 2009;74:264-282. doi:10.1016/j.jvb.2009.01.001
30. Prior RW. Editorial. *J Applied Arts Health* 2009;1:3-6.
31. Chatterjee HJ, Noble G. *Museums, Health and Wellbeing*. Farnham: Ashgate, 2013.
32. Cuypers K, Krokstad S, Holmen TL, Knudtsen MS, Bygren LO, Holmen J. Patterns of
receptive and creative cultural activities and their association with perceived health,
anxiety, depression and satisfaction with life among adults: the HUNT study, Norway. *J*
*Epidemiol Community Health* 2012;66:698-703.
<https://jech.bmj.com/content/jech/66/8/698.full.pdf>
33. Pope C, Mays N. *Qualitative Research in Health Care*, 3rd Edition. Oxford: Blackwell,
2006.
34. Kemmis S, McTaggart R, Nixon R. *The Action Research Planner: Doing Critical*
*Participatory Action Research*. Singapore: Springer 2014.
35. Bradbury H. Introduction: How to Situate and Define Action Research. In H Bradbury
(Ed). *The Sage Handbook of Action Research*, London: Sage 2015. p.1-9.
36. Nicolaidis C, Raymaker DM. Community based participatory research with communities
defined by race, ethnicity, and disability: Translating theory to practice. In H Bradbury
(Ed.), *The Sage Handbook of Action Research*. London: Sage 2015. p.167-178.
37. Daykin N, Stickley T. The role of qualitative research in arts and health. In S Clift, P
Camic (Eds.), *Oxford Textbook of Creative Arts, Health, and Wellbeing: International*
*Perspectives on Practice, Policy and Research*. New York, Oxford: Oxford University
Press 2016, p.73-82.
38. Vaughn LM, Jacquez F. Community-Based Participatory Research Studies Involving
Immigrants. In S Coughlin, SA Smith, ME Fernandez (Eds.), *Handbook of Community-*
*Based Participatory Research*. New York, Oxford: Oxford University Press, 2017. p.115-
130.
39. Blumenthal DS. Is Community-based participatory research possible? *Am J Prev Med*.
2011;40:386-389.
40. Braun V, Clarke V. Using thematic analysis in psychology. *Qual Res Psychol* 2006;3:77-
101.
41. Pope C, Ziebland S, Mays N. *Analysing Qualitative Data*. In C Pope, N Mays. *Qualitative*
*Research in Health Care*, 3rd Edition. Oxford: Blackwell. 2006. p.63-81.
42. Khosravi S. *'Illegal' Traveller: An Auto-ethnography of Borders*. Basingstoke: Palgrave
*MacMillan* 2010.

1
2
3 1 Table 1. Focus group questions

4 Clients:

[revised manuscript text omitted]

Research and reporting methodology

Revised Standards for Quality Improvement Reporting Excellence (SQUIRE 2.0) publication guidelines

Notes to authors

- ▶ The SQUIRE guidelines provide a framework for reporting new knowledge about how to improve healthcare.
- ▶ The SQUIRE guidelines are intended for reports that describe system level work to improve the quality, safety and value of healthcare, and used methods to establish that observed outcomes were due to the intervention(s).
- ▶ A range of approaches exists for improving healthcare. SQUIRE may be adapted for reporting any of these.
- ▶ Authors should consider every SQUIRE item, but it may be inappropriate or unnecessary to include every SQUIRE element in a particular manuscript.
- ▶ The SQUIRE glossary contains definitions of many of the key words in SQUIRE.
- ▶ The explanation and elaboration document provides specific examples of well-written SQUIRE items and an in-depth explanation of each item.
- ▶ Please cite SQUIRE when it is used to write a manuscript.

Text section and item name	Page/line no(s). info is located
Title and abstract	
1. Title	1/1-2/1
Indicate that the manuscript concerns an initiative to improve healthcare (broadly defined to include the quality, safety, effectiveness, patient-centredness, timeliness, cost, efficiency and equity of healthcare).	
Manuscript concerns an initiative to improve healthcare for forcibly displaced people.	
2. Abstract	21/1-9/2
a. Provide adequate information to aid in searching and indexing.	
b. Summarise all key information from various sections of the text using the abstract format of the intended publication or a structured summary such as: background, local problem, methods, interventions, results, conclusions.	
The structure abstract summarises all key information under headings of: objective, design, setting, participants, results, and conclusions.	
Introduction: Why did you start?	
3. Problem description - Nature and significance of the local problem.	2/19-2/22
4. Available knowledge - Summary of what is currently known about the problem, including relevant previous studies.	2/22-2/14
5. Rationale - Informal or formal frameworks, models, concepts and/or theories used to explain the problem, any reasons or assumptions that were used to develop the intervention(s) and reasons why the intervention(s) was expected to work	3/3-3/29
6. Specific aims - Purpose of the project and of this report.	3/30-4/12
Methods: What did you do?	
7. Context - Contextual elements considered important at the outset of introducing the intervention(s).	4/17-4/29

8. Intervention(s)
a. Description of the intervention(s) in sufficient detail that others could reproduce it.	5/26-6/11
b. Specifics of the team involved in the work.	4/32-5/8
9. Study of the intervention(s)
a. Approach chosen for assessing the impact of the intervention(s).	4/17-4/23
b. Approach used to establish whether the observed outcomes were due to the intervention(s).	4/23-4/29
10. Measures
a. Measures chosen for studying processes and outcomes of the intervention(s), including rationale for choosing them, their operational definitions and their validity and reliability.	5/27-5/29
b. Description of the approach to the ongoing assessment of contextual elements that contributed to the success, failure, efficiency and cost.	5/29-6/7
c. Methods employed for assessing completeness and accuracy of data.	6/7-6/11
11. Analysis
a. Qualitative and quantitative methods used to draw inferences from the data.	6/18-6/26
b. Methods for understanding variation within the data, including the effects of time as a variable.	6/26-6/32
12. Ethical considerations - Ethical aspects of implementing and studying the intervention(s) and how they were addressed, including, but not limited to, formal ethics review and potential conflict(s) of interest.	5/14-5/17 13/5-13/8
Results: What did you find?
13. Results
a. Initial steps of the intervention(s) and their evolution over time (eg, time-line diagram, flow chart or table), including modifications made to the intervention during the project.	7/2-7/15 7/22-7/24
b. Details of the process measures and outcomes.	7/17-7/20
7/26-7/28 8/24-8/26 9/26-9/33
c. Contextual elements that interacted with the intervention(s).
25	d. Observed associations between outcomes, interventions and relevant contextual elements.	7/28-10/30
e. Unintended consequences such as unexpected benefits, problems, failures or costs associated with the intervention(s).	10/30-10/34
f. Details about missing data.	11/24-11/27 11/33-12/4
Discussion: What does it mean?
14. Summary
a. Key findings, including relevance to the rationale and specific aims.	11/2-11/7
b. Particular strengths of the project.	11/7-11/13
15. Interpretation
a. Nature of the association between the intervention(s) and the outcomes.	11/14-11/17
b. Comparison of results with findings from other publications.	11/14-11/23
c. Impact of the project on people and systems.	11/10-11/13
38	d. Reasons for any differences between observed and anticipated outcomes, including the influence of context.	12/4-12/7
e. Costs and strategic trade-offs, including opportunity costs.	11/29-11/32

16. Limitations
11/24-11/27
a. Limits to the generalisability of the work.	11/33-12/2
b. Factors that might have limited internal validity such as confounding, bias or imprecision	11/27-11/32
in the design, methods, measurement or analysis.	12/4-12/7
c. Efforts made to minimise and adjust for limitations.	11/7-11/10
12/6-12/7
Conclusions
a. Usefulness of the work.	12/10-12/11
b. Sustainability.	12/11-12/15
c. Potential for spread to other contexts.	12/15-12/18
15	d. Implications for practice and for further study in the field.	12/18-12/21
e. Suggested next steps.	12/21-12/24
Other information
18. Funding - Sources of funding that supported this work. Role, if any, of the funding	13/1-13/3
organisation in the design, implementation, interpretation and reporting.

Ogrinc G, et al. *BMJ Qual Saf* 2015;0:1–7. doi:10.1136/bmjqs-2015-004411

Downloaded from <http://qualitysafety.bmj.com/> on January 2, 2017

BMJ Open

Assessing the impact of artistic and cultural activities on the health and wellbeing of forcibly displaced people using participatory action research

Journal:	BMJ Open
Manuscript ID	bmjopen-2018-025465.R1
Article Type:	Research
Date Submitted by the Author:	01-Oct-2018
Complete List of Authors:	Clini, Clelia; Loughborough University - London, Institute for Media and Creative Industries Thomson, Linda; University College London Division of Biosciences Chatterjee, Helen; University College London, Division of Biosciences
Primary Subject Heading:	Qualitative research
Secondary Subject Heading:	Public health, Mental health
Keywords:	creative activities, forced displacement, participatory action research, post-traumatic stress disorder, refugees and asylum seekers

Assessing the impact of artistic and cultural activities on the health and wellbeing of forcibly
displaced people using participatory action research

Clelia Clini, Linda J Thomson, Helen J Chatterjee

Correspondence to: Helen J Chatterjee, UCL Division of Biosciences, 507B Darwin
Building, University College London, London, WC1E 6BT, UK. email
8 h.chatterjee@ucl.ac.uk telephone +44(0)2031084104 fax +44(0)2076797193

Clelia Clini, Institute for Media and Creative Industries, Loughborough University London,
UK

Linda J Thomson, UCL Division of Biosciences, University College London, London, UK

Key words: creative activities; forced displacement; participatory action research; post-
traumatic stress disorder; refugees and asylum seekers;

Word count: 5080

**Abstract**

Objective: Drawing upon a growing body of research suggesting that taking part in artistic
and cultural activities benefits health and wellbeing, the objective was to develop a
participatory action research (PAR) method for assessing the impact of arts interventions on
forcibly displaced people, and identify themes concerning perceived benefits of such
programmes.

Design: A collaborative study following PAR principles of observation, focus groups and in-
depth semi-structured interviews.

Setting. London-based charity working with asylum seekers and refugees.

Participants: An opportunity sample (n=31; 6m) participated in focus groups comprising
refugees/asylum seekers (n=12; 2m), volunteers (n=4; 1m) and charity staff (n=15, 3m). A
subset of these (n=17; 3m) participated in interviews comprising refugees/asylum seekers
(n=7; 1m), volunteers (n=7; 1 m) and charity staff (n=3; 1m).

[revised manuscript text omitted]

PAR was an appropriate method to use for collaboration between displaced communities and
academics. The PAR approach was particularly useful when conducting research with
refugees/asylum seekers suffering from PTSD and other mental health issues because the
process of involvement as co-researchers helped to counter client isolation and build social
capital through regular meetings. In helping to interpret findings, clients acquired research
methods skills (e.g. drafting interview guides, conducting focus groups, discussing data) and
organisational skills (event/exhibition). Participating in the research and disseminating
findings gave clients a sense of agency and ownership of the project that enhanced self-
confidence and feelings of empowerment and provided a positive means of expression
countering their refugee/asylum seeker status that effectively placed them powerless.

[revised manuscript text omitted]

**References**

- 1. All-Party Parliamentary Group on Arts, Health and Wellbeing (APPGAHW). Creative
Health: The Arts for Health and Wellbeing. Inquiry Report. 2017;
<http://www.artshealthandwellbeing.org.uk/appg-inquiry/>
2. New Economics Foundation (NEF). 2009. National accounts of wellbeing: What is
wellbeing? NEF. [http://www.nationalaccountsofwellbeing.org/learn/what-is-well-](http://www.nationalaccountsofwellbeing.org/learn/what-is-well-being.html)
[being.html](http://www.nationalaccountsofwellbeing.org/learn/what-is-well-being.html)
3. New Economic Foundation (NEF). National Accounts of Well-being: Bringing Real
Wealth onto the Balance Sheet. 2009.
https://neweconomics.org/uploads/files/2027fb05fed1554aea_uim6vd4c5.pdf
4. Ander E, Thomson LJ, Noble G, Lanceley A, Menon U, Chatterjee H. Generic well-being
outcomes: towards a conceptual framework for well-being outcomes in museums.
Museum Management Curatorship 2011; 26:237–259.
doi:10.1080/09647775.2011.585798
5. Crawley H, Düvell F, Jones K, MCMahon s, Sigona, N. Destination Europe?
Understanding the dynamics and drivers of Mediterranean migration in 2015. MEDMIG
Final Report 2016. www.medmig.info/research-brief-destination-europe.pdf
6. Vaughan-Williams N. “We are *not* animals!” Humanitarian border security and
zoopolitical spaces in Europe. Political Geography 2015;45:1–10.
doi:10.1016/j.polgeo.2014.09.009

- 7. Katona C. Non-affective psychosis in refugees. *Br Med J* 2016;352:i1279.
doi:10.1136/bmj.i1279
- 8. Robjant K, Robbins I, Senior V. Psychological distress amongst immigration detainees: A
cross-sectional questionnaire study. *Brit J Clin Psych* 2009;48:275–286.
doi:10.1348/014466508X397007
- 9. Alpak G, Unal A, Bulbul F, Sagaltici E, Bez Y, Altindag A et al. Post-traumatic stress
disorder among Syrian refugees in Turkey: A cross-sectional study. *Int J Psych Clinic*
*Practice* 2015;19:45–50. doi:10.3109/13651501.2014.961930
- 10. Schubert CC, Punamäki R. Mental health among torture survivors: cultural background,
refugee status and gender. *Nordic J Psych* 2011;65:175–182.
doi:10.3109/08039488.2010.514943
- 11. Silove D, Sinnerbrink I, Field A, Manicavasagar V, Steel Z. Anxiety, depression and
PTSD in asylum-seekers: associations with pre-migration trauma and post-migration
stressors. *Brit J Psych* 1997;170:351–357. doi:10.1192/bjp.170.4.351
doi:10.1348/014466508X397007
- 12. Aragona M, Pucci D, Mazzetti M, Geraci S. Post-migration living difficulties as a
significant risk factor for PTSD in immigrants: a primary care study. *Italian J Public*
*Health* 2012;9:e7525;1–8. doi:10.2427/7525
- 13. Allen J, Allen M. The social determinants of health, empowerment and participation. In S
Clift, P Camic (Eds.), *Oxford Textbook of Creative Arts, Health, and Wellbeing:*
*International Perspectives on Practice, Policy and Research.* New York, Oxford: Oxford
University Press; 2016, p27–34.
- 14. Fisher M. Why mental health is a political issue. *The Guardian*, July 16, 2012.
<https://www.theguardian.com/commentisfree/2012/jul/16/mental-health-political-issue>
- 15. Griffin J. *The Lonely Society.* London: The Mental Health Foundation.
https://www.mentalhealth.org.uk/sites/default/files/the_lonely_society_report.pdf
- 16. Marmot M, Allen J, Goldblatt P, Boyce T, McNeish D, Grady M et al. *Fair Society,*
*Healthy Lives: The Marmot Review.* 2010;
[http://www.instituteofhealthequity.org/resources-reports/fair-society-healthy-lives-the-](http://www.instituteofhealthequity.org/resources-reports/fair-society-healthy-lives-the-marmot-review/fair-society-healthy-lives-full-report-pdf)
[marmot-review/fair-society-healthy-lives-full-report-pdf](http://www.instituteofhealthequity.org/resources-reports/fair-society-healthy-lives-the-marmot-review/fair-society-healthy-lives-full-report-pdf)
- 17. Paul K, Moser K. Unemployment impairs mental health: Meta-analyses. *J Vocational*
*Behavior* 2009;74:264-282. doi:10.1016/j.jvb.2009.01.001
- 18. Prior RW. Editorial. *J Applied Arts Health* 2009;1:3–6.

19. Bygren LO, Johansson S, Konlaan BB, Grijbovski AM, Wilkinson AV, Sjöström M.
Attending cultural events and cancer mortality: A Swedish cohort study. *Arts Health: Int J*
*Res Pol Pract* 2009;1:64–73. doi:10.1080/17533010802528058
20. Camic P, Chatterjee H. Museums and art galleries as partners for public health
interventions. *Perspect Public Health* 2013;133:66–77.
21. Chatterjee H. Museums and art galleries as settings for public health interventions. In S
Clift, P Camic (Eds.), *Oxford Textbook of Creative Arts, Health, and Wellbeing:*
*International Perspectives on Practice, Policy and Research*. New York, Oxford: Oxford
University Press 2016; p.281–289.
22. Clift S, Skingley A, Coulton S, Rodriguez J. The effectiveness and cost-effectiveness of a
participative community singing programme as a health promotion initiative for older
people: protocol for a randomised controlled trial. *BMC Public Health* 2011;11:1–6.
doi:10.1186/1471-2458-11-142
23. Clift S, Camic P. *Oxford Textbook of Creative Arts, Health, and Wellbeing: International*
*Perspectives on Practice, Policy and Research*. New York, Oxford: Oxford University
Press, 2016.
24. Konlaan BB, Bygren LO, Johansson S. Visiting the cinema, concerts, museums or art
exhibitions as determinant of survival: a Swedish fourteen-year cohort follow-up. *Scand J*
*Public Health* 2000;28:174–178. doi:10.1177/14034948000280030501
25. Staricoff RL. *Arts in health: A review of the medical literature*. London: Arts Council
England, Research Report 2004;36:1–91. [http://www.artsandhealth.ie/wp-](http://www.artsandhealth.ie/wp-content/uploads/2011/08/AHReview-of-Medical-Literature1.pdf)
[content/uploads/2011/08/AHReview-of-Medical-Literature1.pdf](http://www.artsandhealth.ie/wp-content/uploads/2011/08/AHReview-of-Medical-Literature1.pdf)
26. Staricoff RL. *Arts in health: The value of evaluation*. *Perspect Public Health*
2006;126:116–120. <http://journals.sagepub.com/doi/pdf/10.1177/1466424006064300>
27. Chatterjee HJ, Noble G. *Museums, Health and Wellbeing*. Farnham: Ashgate, 2013.
28. Cuypers K, Krokstad S, Holmen TL, Knudtsen MS, Bygren LO, Holmen J. Patterns of
receptive and creative cultural activities and their association with perceived health,
anxiety, depression and satisfaction with life among adults: the HUNT study, Norway. *J*
*Epidemiol Community Health* 2012;66:698–703.
<https://jech.bmj.com/content/jech/66/8/698.full.pdf>
29. Kidd B, Zahir S, Khan S. *Arts and Refugees: History, Impact and Future*. London: Arts
Council England, 2008. [https://baringfoundation.org.uk/wp-](https://baringfoundation.org.uk/wp-content/uploads/2014/10/ArtsandRefugees.pdf)
[content/uploads/2014/10/ArtsandRefugees.pdf](https://baringfoundation.org.uk/wp-content/uploads/2014/10/ArtsandRefugees.pdf)

30. Kemmis S, McTaggart R, Nixon R. *The Action Research Planner: Doing Critical*
*Participatory Action Research*. Singapore: Springer 2014.
31. Bradbury H. Introduction: How to Situate and Define Action Research. In H Bradbury
(Ed). *The Sage Handbook of Action Research*, London: Sage 2015. p.1–9.
32. Nicolaidis C, Raymaker DM. Community based participatory research with communities
defined by race, ethnicity, and disability: Translating theory to practice. In H Bradbury
(Ed.), *The Sage Handbook of Action Research*. London: Sage 2015. p.167–178.
33. Daykin N, Stickley T. The role of qualitative research in arts and health. In S Clift, P
Camic (Eds.), *Oxford Textbook of Creative Arts, Health, and Wellbeing: International*
*Perspectives on Practice, Policy and Research*. New York, Oxford: Oxford University
Press 2016, p.73–82.
34. Vaughn LM, Jacquez F. Community-Based Participatory Research Studies Involving
Immigrants. In S Coughlin, SA Smith, ME Fernandez (Eds.), *Handbook of Community-*
*Based Participatory Research*. New York, Oxford: Oxford University Press, 2017. p.115–
130.
35. Blumenthal DS. Is Community-based participatory research possible? *Am J Prev Med*.
2011;40:386–389.
36. Braun V, Clarke V. Using thematic analysis in psychology. *Qual Res Psychol* 2006;3:77–
101.
37. Pope C, Mays N. *Qualitative Research in Health Care*, 3rd Edition. Oxford: Blackwell,
2006.
38. Pope C, Ziebland S, Mays N. Analysing Qualitative Data. In C Pope, N Mays. *Qualitative*
*Research in Health Care*, 3rd Edition. Oxford: Blackwell. 2006. p.63–81.

[revised manuscript text omitted]

-

Research and reporting methodology

Revised Standards for Quality Improvement Reporting Excellence (SQIRE 2.0) publication guidelines

Notes to authors

- ▶ The SQIRE guidelines provide a framework for reporting new knowledge about how to improve healthcare.
- ▶ The SQIRE guidelines are intended for reports that describe system level work to improve the quality, safety and value of healthcare, and used methods to establish that observed outcomes were due to the intervention(s).
- ▶ A range of approaches exists for improving healthcare. SQIRE may be adapted for reporting any of these.
- ▶ Authors should consider every SQIRE item, but it may be inappropriate or unnecessary to include every SQIRE element in a particular manuscript.
- ▶ The SQIRE glossary contains definitions of many of the key words in SQIRE.
- ▶ The explanation and elaboration document provides specific examples of well-written SQIRE items and an in-depth explanation of each item.
- ▶ Please cite SQIRE when it is used to write a manuscript.

Text section and item name	Page/line no(s). info is located
Title and abstract	
1. Title	1/1-2
Indicate that the manuscript concerns an initiative to improve healthcare (broadly defined to include the quality, safety, effectiveness, patient-centredness, timeliness, cost, efficiency and equity of healthcare).	
Response: Manuscript concerns an initiative to improve healthcare for forcibly displaced people.	Yes (explained left)
2. Abstract	1/1 - 2/9
a. Provide adequate information to aid in searching and indexing.	Yes
b. Summarise all key information from various sections of the text using the abstract format of the intended publication or a structured summary such as: background, local problem, methods, interventions, results, conclusions.	
Response: The abstract summarises all key information under headings of: objective, design, setting, participants, results, and conclusions.	Yes (explained left)
Introduction: Why did you start?	
3. Problem description - Nature and significance of the local problem.	2/32 - 3/8
4. Available knowledge - Summary of what is currently known about the problem, including relevant previous studies.	3/9-27
5. Rationale - Informal or formal frameworks, models, concepts and/or theories used to explain the problem, any reasons or assumptions that were used to develop the intervention(s) and reasons why the intervention(s) was expected to work	3/28 - 4/11
6. Specific aims - Purpose of the project and of this report.	4/12-33
Methods: What did you do?	
7. Context - Contextual elements considered important at the outset of introducing the intervention(s).	5/8-21

8. Intervention(s)
a. Description of the intervention(s) in sufficient detail that others could reproduce it.	6/10-27
b. Specifics of the team involved in the work.	5/23 – 6/8
9. Study of the intervention(s)
a. Approach chosen for assessing the impact of the intervention(s).	5/3-6
b. Approach used to establish whether the observed outcomes were due to the
intervention(s).	6/24-29
10. Measures
a. Measures chosen for studying processes and outcomes of the intervention(s), including
rationale for choosing them, their operational definitions and their validity and reliability.	5/23 – 6/8
b. Description of the approach to the ongoing assessment of contextual elements that
contributed to the success, failure, efficiency and cost.	6/10-24
c. Methods employed for assessing completeness and accuracy of data.	6/24-29
11. Analysis
a. Qualitative and quantitative methods used to draw inferences from the data.	7/1-17
b. Methods for understanding variation within the data, including the effects of time as a
variable.	6/11-13 and
7/9-13
12. Ethical considerations - Ethical aspects of implementing and studying the intervention(s)
and how they were addressed, including, but not limited to, formal ethics review and
potential conflict(s) of interest.	15/1-5
Results: What did you find?
13. Results
a. Initial steps of the intervention(s) and their evolution over time (e.g., time-line diagram,
flow chart or table), including modifications made to the intervention during the project.	6/11-13
b. Details of the process measures and outcomes.	7/21-26 and
8/22-23
c. Contextual elements that interacted with the intervention(s).	7/26 – 8/19 and
8/25 – 11/28
34	d. Observed associations between outcomes, interventions and relevant contextual	
elements.	9/14-18; 9/29-
33; 10/8-12;
11/9-11 and
11/16-18
e. Unintended consequences such as unexpected benefits, problems, failures or costs
associated with the intervention(s).	11/28-32 and
9/17-18
f. Details about missing data.	12/13-15 and
12/34 – 13/2
Discussion: What does it mean?
14. Summary
a. Key findings, including relevance to the rationale and specific aims.	12/2-8; 12/18-
23; 12/26-29
and 12/31 -13/2
b. Particular strengths of the project.	12/8-13; 12/15-
17; 13/3-7 and
13/12-15
15. Interpretation
a. Nature of the association between the intervention(s) and the outcomes.	13/7-15

1	b. Comparison of results with findings from other publications.	12/19-31 and 14/7-10
2		
3	c. Impact of the project on people and systems.	13/12-15 and 14/14-22
4		
5	d. Reasons for any differences between observed and anticipated outcomes, including the 6 influence of context.	13/16-30 and 13/30-33
e. Costs and strategic trade-offs, including opportunity costs.	14/7-14
16. Limitations
a. Limits to the generalisability of the work.	14/14-20
b. Factors that might have limited internal validity such as confounding, bias or imprecision 14 in the design, methods, measurement or analysis.	13/16-17; 13/22-25 and 13/27-30
c. Efforts made to minimise and adjust for limitations.	13/30-33
Conclusions
a. Usefulness of the work.	13/3-10
b. Sustainability.	13/10-15
c. Potential for spread to other contexts.	14/14-17
23	d. Implications for practice and for further study in the field.	14/17-20
e. Suggested next steps.	14/20-22
Other information
18. Funding - Sources of funding that supported this work. Role, if any, of the funding 30 organisation in the design, implementation, interpretation and reporting.	14/31-33

Ogrinc G, et al. *BMJ Qual Saf* 2015;0:1–7. doi:10.1136/bmjqs-2015-004411

Downloaded from <http://qualitysafety.bmj.com/> on January 2, 2017

BMJ Open

Assessing the impact of artistic and cultural activities on the health and wellbeing of forcibly displaced people using participatory action research

Journal:	BMJ Open
Manuscript ID	bmjopen-2018-025465.R2
Article Type:	Research
Date Submitted by the Author:	17-Oct-2018
Complete List of Authors:	Clini, Clelia; Loughborough University - London, Institute for Media and Creative Industries Thomson, Linda; University College London Division of Biosciences Chatterjee, Helen; University College London, Division of Biosciences
Primary Subject Heading:	Qualitative research
Secondary Subject Heading:	Public health, Mental health
Keywords:	creative activities, forced displacement, participatory action research, post-traumatic stress disorder, refugees and asylum seekers

Assessing the impact of artistic and cultural activities on the health and wellbeing of forcibly
displaced people using participatory action research

Clelia Clini, Linda J Thomson, Helen J Chatterjee

Correspondence to: Helen J Chatterjee, UCL Division of Biosciences, 507B Darwin Building,
University College London, London, WC1E 6BT, UK.

email h.chatterjee@ucl.ac.uk telephone +44(0)2031084104 fax +44(0)2076797193

Clelia Clini, Institute for Media and Creative Industries, Loughborough University London, UK

Linda J Thomson, UCL Division of Biosciences, University College London, London, UK

Key words: creative activities; forced displacement; participatory action research; post-traumatic
stress disorder; refugees and asylum seekers;

Word count: 5315

24 19 **Abstract**

[revised manuscript text omitted]

Results

Focus Groups

Focus group data were analysed using thematic analysis (NVivo11). Findings showed consistency in the way all HBF participants (clients, volunteers and staff) articulated their reflections on the impact of creative activities. The benefits of creative activities were highlighted in clusters emerging from around three main overarching themes: ‘social aspects/friendships’, ‘skills’, and ‘mood-personal sphere’; the latter sub-divided into ‘brain’ (creative thought processes), ‘routine’, self-expression’ and ‘confidence’. During focus groups, co-researchers were asked to write answers to the questions posed, and a collective discussion followed. Clients responded consistently about benefits of the creative programmes. One client listed ‘Activities help my brain think about other stuff and keep me busy and relax. You can’t think about bad things. To learn more skills. I can use skills to help others.’ Staff commented on the mood enhancing benefits of CAP for people with PTSD; one staff member observed ‘creative activities allow clients to momentarily leave out their problems or memories, by focusing on a particular activity they cannot think about anything else’. The focus group with HFB staff emphasised that being busy in the company of others allowed clients to feel safe and not to dwell upon their own situation. In concentrating on learning new skills, clients blocked out (albeit for short periods) memories of the past. Learning new or improving existing skills, the second theme to emerge during focus groups, appeared to be an important factor in improving self-esteem; another client noted that 
[revised manuscript text omitted]

Museum Manage Curator 2011; 26:237–259. doi:10.1080/09647775.2011.585798
- 5. Crawley H, Düvell F, Jones K, McMahon S, Sigona, N. Destination Europe? Understanding
the dynamics and drivers of Mediterranean migration in 2015. MEDMIG Final Report 2016.
www.medmig.info/research-brief-destination-europe.pdf
- 6. Vaughan-Williams N. “We are *not* animals!” Humanitarian border security and zoopolitical
spaces in Europe. Political Geography 2015;45:1–10. doi:10.1016/j.polgeo.2014.09.009
- 7. Katona C. Non-affective psychosis in refugees. Br Med J 2016;352:i1279.
doi:10.1136/bmj.i1279
- 8. Robjant K, Robbins I, Senior V. Psychological distress amongst immigration detainees: A
cross-sectional questionnaire study. Br J Clin Psychol 2009;48:275–286.
doi:10.1348/014466508X397007
- 9. Alpak G, Unal A, Bulbul F, Sagaltici E, Bez Y, Altindag A et al. Post-traumatic stress
disorder among Syrian refugees in Turkey: A cross-sectional study. Int J Psychol Clinic
Practice 2015;19:45–50. doi:10.3109/13651501.2014.961930
- 10. Schubert CC, Punamäki R. Mental health among torture survivors: cultural background,
refugee status and gender. Nordic J Psychol 2011;65:175–182.
doi:10.3109/08039488.2010.514943
- 11. Silove D, Sinnerbrink I, Field A, Manicavasagar V, Steel Z. Anxiety, depression and PTSD in
asylum-seekers: associations with pre-migration trauma and post-migration stressors. Brit J
Psych 1997;170:351–357. doi:10.1192/bjp.170.4.351
doi:10.1348/014466508X397007

12. Aragona M, Pucci D, Mazzetti M, Geraci S. Post-migration living difficulties as a significant
risk factor for PTSD in immigrants: a primary care study. *Italian J Public Health*
2012;9:e7525;1–8. doi:10.2427/7525
- 13. Allen J, Allen M. The social determinants of health, empowerment and participation. In S
Clift, P Camic (Eds.), *Oxford Textbook of Creative Arts, Health, and Wellbeing: International*
*Perspectives on Practice, Policy and Research*. New York, Oxford: Oxford University Press;
2016, p27–34.
- 14. Fisher M. Why mental health is a political issue. *The Guardian*, July 16, 2012.
<https://www.theguardian.com/commentisfree/2012/jul/16/mental-health-political-issue>
- 15. Griffin J. *The Lonely Society*. London: The Mental Health Foundation.
https://www.mentalhealth.org.uk/sites/default/files/the_lonely_society_report.pdf
- 16. Marmot M, Allen J, Goldblatt P, Boyce T, McNeish D, Grady M et al. *Fair Society, Healthy*
*Lives: The Marmot Review*. 2010; [http://www.instituteofhealthequity.org/resources-](http://www.instituteofhealthequity.org/resources-reports/fair-society-healthy-lives-the-marmot-review/fair-society-healthy-lives-full-report-pdf)
[reports/fair-society-healthy-lives-the-marmot-review/fair-society-healthy-lives-full-report-pdf](http://www.instituteofhealthequity.org/resources-reports/fair-society-healthy-lives-the-marmot-review/fair-society-healthy-lives-full-report-pdf)
- 17. Paul K, Moser K. Unemployment impairs mental health: Meta-analyses. *J Vocational*
*Behavior* 2009;74:264-282. doi:10.1016/j.jvb.2009.01.001
- 18. Prior RW. Editorial. *J Applied Arts Health* 2009;1:3–6.
- 19. Bygren LO, Johansson S, Konlaan BB, Grjibovski AM, Wilkinson AV, Sjöström M.
*Attending cultural events and cancer mortality: A Swedish cohort study*. *Arts Health: Int J*
*Res Pol Pract* 2009;1:64–73. doi:10.1080/17533010802528058
- 20. Camic P, Chatterjee H. Museums and art galleries as partners for public health interventions.
*Perspect Public Health* 2013;133:66–77.
- 21. Chatterjee H. Museums and art galleries as settings for public health interventions. In S Clift,
P Camic (Eds.), *Oxford Textbook of Creative Arts, Health, and Wellbeing: International*
*Perspectives on Practice, Policy and Research*. New York, Oxford: Oxford University Press
2016; p.281–289.
- 22. Clift S, Skingley A, Coulton S, Rodriguez J. The effectiveness and cost-effectiveness of a
participative community singing programme as a health promotion initiative for older people:
protocol for a randomised controlled trial. *BMC Public Health* 2011;11:1–6.
doi:10.1186/1471-2458-11-142

- 23. Clift S, Camic P. Oxford Textbook of Creative Arts, Health, and Wellbeing: International
Perspectives on Practice, Policy and Research. New York, Oxford: Oxford University Press,
2016.
- 24. Konlaan BB, Bygren LO, Johansson S. Visiting the cinema, concerts, museums or art
exhibitions as determinant of survival: a Swedish fourteen-year cohort follow-up. Scand J
Public Health 2000;28:174–178. doi:10.1177/14034948000280030501
- 25. Staricoff RL. Arts in health: A review of the medical literature. London: Arts Council
England, Research Report 2004;36:1–91. [http://www.artsandhealth.ie/wp-](http://www.artsandhealth.ie/wp-content/uploads/2011/08/AHReview-of-Medical-Literature1.pdf)
[content/uploads/2011/08/AHReview-of-Medical-Literature1.pdf](http://www.artsandhealth.ie/wp-content/uploads/2011/08/AHReview-of-Medical-Literature1.pdf)
- 26. Staricoff RL. Arts in health: The value of evaluation. Perspect Public Health 2006;126:116–
120. <http://journals.sagepub.com/doi/pdf/10.1177/1466424006064300>
- 27. Chatterjee HJ, Noble G. Museums, Health and Wellbeing. Farnham: Ashgate, 2013.
- 28. Cuypers K, Krokstad S, Holmen TL, Knudtsen MS, Bygren LO, Holmen J. Patterns of
receptive and creative cultural activities and their association with perceived health, anxiety,
depression and satisfaction with life among adults: the HUNT study, Norway. J Epidemiol
Community Health 2012;66:698–703. <https://jech.bmj.com/content/jech/66/8/698.full.pdf>
- 29. Kidd B, Zahir S, Khan S. Arts and Refugees: History, Impact and Future. London: Arts
Council England, 2008. [https://baringfoundation.org.uk/wp-](https://baringfoundation.org.uk/wp-content/uploads/2014/10/ArtsandRefugees.pdf)
[content/uploads/2014/10/ArtsandRefugees.pdf](https://baringfoundation.org.uk/wp-content/uploads/2014/10/ArtsandRefugees.pdf)
- 30. Kemmis S, McTaggart R, Nixon R. The Action Research Planner: Doing Critical
Participatory Action Research. Singapore: Springer 2014.
- 31. Bradbury H. Introduction: How to situate and define action research. In H Bradbury (Ed.).
The Sage Handbook of Action Research, London: Sage 2015. p.1–9.
- 32. Nicolaidis C, Raymaker DM. Community based participatory research with communities
defined by race, ethnicity, and disability: Translating theory to practice. In H Bradbury (Ed.),
The Sage Handbook of Action Research. London: Sage 2015. p.167–178.
- 33. Daykin N, Stickley T. The role of qualitative research in arts and health. In S Clift, P Camic
(Eds.), Oxford Textbook of Creative Arts, Health, and Wellbeing: International Perspectives
on Practice, Policy and Research. New York, Oxford: Oxford University Press 2016, p.73–82.

1 34. Vaughn LM, Jacquez F. Community-Based Participatory Research Studies Involving
2 Immigrants. In S Coughlin, SA Smith, ME Fernandez (Eds.), Handbook of Community-Based
3 Participatory Research. New York, Oxford: Oxford University Press, 2017. p.115–130.
35. Blumenthal DS. Is Community-based participatory research possible? Am J Prev Med.
4 2011;40:386–389.
36. Braun V, Clarke V. Using thematic analysis in psychology. Qual Res Psychol 2006;3:77–101.
37. Pope C, Mays N. Qualitative Research in Health Care, 3rd Edition. Oxford: Blackwell, 2006.
38. Pope C, Ziebland S, Mays N. Analysing Qualitative Data. In C Pope, N Mays. Qualitative
8 Research in Health Care, 3rd Edition. Oxford: Blackwell. 2006. p.63–81.

[revised manuscript text omitted]

For peer review only

Research and reporting methodology

Revised **Standards for Quality Improvement Reporting Excellence (SQIRE 2.0)** publication guidelines

Notes to authors

- ▶ The SQIRE guidelines provide a framework for reporting new knowledge about how to improve healthcare.
- ▶ The SQIRE guidelines are intended for reports that describe system level work to improve the quality, safety and value of healthcare, and used methods to establish that observed outcomes were due to the intervention(s).
- ▶ A range of approaches exists for improving healthcare. SQIRE may be adapted for reporting any of these.
- ▶ Authors should consider every SQIRE item, but it may be inappropriate or unnecessary to include every SQIRE element in a particular manuscript.
- ▶ The SQIRE glossary contains definitions of many of the key words in SQIRE.
- ▶ The explanation and elaboration document provides specific examples of well-written SQIRE items and an in-depth explanation of each item.
- ▶ Please cite SQIRE when it is used to write a manuscript.

Text section and item name	Page/line no(s). info is located
Title and abstract	
1. Title	1/1-1/2
Indicate that the manuscript concerns an initiative to improve healthcare (broadly defined to include the quality, safety, effectiveness, patient-centredness, timeliness, cost, efficiency and equity of healthcare).	
Manuscript concerns an initiative to improve healthcare for forcibly displaced people.	
2. Abstract	1/21-2/9
a. Provide adequate information to aid in searching and indexing.	1/21-2/9
b. Summarise all key information from various sections of the text using the abstract format of the intended publication or a structured summary such as: background, local problem, methods, interventions, results, conclusions.	1/21-2/9
The structure abstract summarises all key information under headings of: objective, design, setting, participants, results, and conclusions.	1/21-2/9
Introduction: Why did you start?	
3. Problem description - Nature and significance of the local problem.	2/32-3/8 and 3/28-4/11
4. Available knowledge - Summary of what is currently known about the problem, including relevant previous studies.	3/9-3/27
5. Rationale - Informal or formal frameworks, models, concepts and/or theories used to explain the problem, any reasons or assumptions that were used to develop the intervention(s) and reasons why the intervention(s) was expected to work	2/19-2/31 and 4/12-4/24
6. Specific aims - Purpose of the project and of this report.	4/25-4/33
Methods: What did you do?	
7. Context - Contextual elements considered important at the outset of introducing the intervention(s).	5/4-5/6

8. Intervention(s)
a. Description of the intervention(s) in sufficient detail that others could reproduce it.	5/9-5/21
b. Specifics of the team involved in the work.	5/24-5/27
9. Study of the intervention(s)
a. Approach chosen for assessing the impact of the intervention(s).	5/27-6/8 6/11-6/23
b. Approach used to establish whether the observed outcomes were due to the intervention(s).	6/23-6/27
10. Measures
a. Measures chosen for studying processes and outcomes of the intervention(s), including rationale for choosing them, their operational definitions and their validity and reliability.	6/27-6/29
b. Description of the approach to the ongoing assessment of contextual elements that contributed to the success, failure, efficiency and cost.	7/2-7/11
c. Methods employed for assessing completeness and accuracy of data.	7/11-7/17
11. Analysis
a. Qualitative and quantitative methods used to draw inferences from the data.	7/2-7/10
b. Methods for understanding variation within the data, including the effects of time as a variable.	6/14-6/17 and 7/10-7/17
12. Ethical considerations - Ethical aspects of implementing and studying the intervention(s) and how they were addressed, including, but not limited to, formal ethics review and potential conflict(s) of interest.	15/2-15/5
Results: What did you find?
13. Results
a. Initial steps of the intervention(s) and their evolution over time (eg, time-line diagram, flow chart or table), including modifications made to the intervention during the project.	5/3-5/21
b. Details of the process measures and outcomes.	6/10-7/17
c. Contextual elements that interacted with the intervention(s).	5/23-5/31 and 5/34-6/8
36	d. Observed associations between outcomes, interventions and relevant contextual elements.	7/19-7/33, 8/2-8/5, 8/10-8/21 and 8/26-11/32
e. Unintended consequences such as unexpected benefits, problems, failures or costs associated with the intervention(s).	7/33-8/2, 8/6-8/9 and 8/32-8/34
f. Details about missing data.	5/31-5/34
Discussion: What does it mean?
14. Summary
a. Key findings, including relevance to the rationale and specific aims.	12/2-12/8
b. Particular strengths of the project.	12/8-12/13 and 12/15-12/17
15. Interpretation
a. Nature of the association between the intervention(s) and the outcomes.	12/18-12/23
b. Comparison of results with findings from other publications.	12/23-12/31

1	c. Impact of the project on people and systems.	13/3-13/17
2	d. Reasons for any differences between observed and anticipated outcomes, including the	12/13-12/15
influence of context.	and 12/31-13/2
e. Costs and strategic trade-offs, including opportunity costs.	13/17-13/27
16. Limitations
a. Limits to the generalisability of the work.	13/28-13/31
b. Factors that might have limited internal validity such as confounding, bias or imprecision	14/32-14/8
in the design, methods, measurement or analysis.
c. Efforts made to minimise and adjust for limitations.	14/8-14/11
Conclusions
a. Usefulness of the work.	14/14-14/19
b. Sustainability.	14/19-14/22
c. Potential for spread to other contexts.	14/22-14/26
18	d. Implications for practice and for further study in the field.	14/26-14/29
e. Suggested next steps.	14/29-14/34
Other information
18. Funding - Sources of funding that supported this work. Role, if any, of the funding	15/9-15/11
organisation in the design, implementation, interpretation and reporting.

Ogrinc G, et al. *BMJ Qual Saf* 2015;0:1–7. doi:10.1136/bmjqs-2015-004411

Downloaded from <http://qualitysafety.bmj.com/> on January 2, 2017

BMJ Open

Assessing the impact of artistic and cultural activities on the health and wellbeing of forcibly displaced people using participatory action research

Journal:	BMJ Open
Manuscript ID	bmjopen-2018-025465.R3
Article Type:	Research
Date Submitted by the Author:	14-Dec-2018
Complete List of Authors:	Clini, Clelia; Loughborough University - London, Institute for Media and Creative Industries Thomson, Linda; University College London Division of Biosciences Chatterjee, Helen; University College London, Division of Biosciences
Primary Subject Heading:	Qualitative research
Secondary Subject Heading:	Public health, Mental health
Keywords:	creative activities, forced displacement, participatory action research, post-traumatic stress disorder, refugees and asylum seekers

1 Assessing the impact of artistic and cultural activities on the health and wellbeing of forcibly
displaced people using participatory action research

Clelia Clini, Linda J Thomson, Helen J Chatterjee

Correspondence to: Helen J Chatterjee, UCL Division of Biosciences, 507B Darwin
Building, University College London, London, WC1E 6BT, UK.
email h.chatterjee@ucl.ac.uk telephone +44(0)2031084104 fax +44(0)2076797193

Clelia Clini, Institute for Media and Creative Industries, Loughborough University London,
UK

Linda J Thomson, UCL Division of Biosciences, University College London, London, UK

Key words: creative activities; forced displacement; participatory action research; post-
traumatic stress disorder; refugees and asylum seekers;

Word count: 5342

34 20 **Abstract**

[revised manuscript text omitted]
5. Crawley H, Düvell F, Jones K, McMahon s, Sigona, N. Destination Europe? Understanding the dynamics and drivers of Mediterranean migration in 2015. MEDMIG Final Report 2016. www.medmig.info/research-brief-destination-europe.pdf
6. Vaughan-Williams N. “We are *not* animals!” Humanitarian border security and zoopolitical spaces in Europe. *Political Geography* 2015;45:1–10. doi:10.1016/j.polgeo.2014.09.009
7. Katona C. Non-affective psychosis in refugees. *Br Med J* 2016;352:i1279. doi:10.1136/bmj.i1279
8. Robjant K, Robbins I, Senior V. Psychological distress amongst immigration detainees: A cross-sectional questionnaire study. *Br J Clin Psychol* 2009;48:275–286. doi:10.1348/014466508X397007
9. Alpak G, Unal A, Bulbul F, Sagaltici E, Bez Y, Altindag A et al. Post-traumatic stress disorder among Syrian refugees in Turkey: A cross-sectional study. *Int J Psychol Clinic Practice* 2015;19:45–50. doi:10.3109/13651501.2014.961930
10. Schubert CC, Punamäki R. Mental health among torture survivors: cultural background, refugee status and gender. *Nordic J Psychol* 2011;65:175–182. doi:10.3109/08039488.2010.514943
11. Silove D, Sinnerbrink I, Field A, Manicavasagar V, Steel Z. Anxiety, depression and PTSD in asylum-seekers: associations with pre-migration trauma and post-migration stressors. *Brit J Psych* 1997;170:351–357. doi:10.1192/bjp.170.4.351

doi:10.1348/014466508X397007
12. Aragona M, Pucci D, Mazzetti M, Geraci S. Post-migration living difficulties as a
significant risk factor for PTSD in immigrants: a primary care study. *Italian J Public*
*Health* 2012;9:e7525;1–8. doi:10.2427/7525
13. Allen J, Allen M. The social determinants of health, empowerment and participation. In S
Clift, P Camic (Eds.), *Oxford Textbook of Creative Arts, Health, and Wellbeing:*
*International Perspectives on Practice, Policy and Research*. New York, Oxford: Oxford
University Press; 2016, p27–34.
14. Fisher M. Why mental health is a political issue. *The Guardian*, July 16, 2012.
<https://www.theguardian.com/commentisfree/2012/jul/16/mental-health-political-issue>
15. Griffin J. *The Lonely Society*. London: The Mental Health Foundation.
https://www.mentalhealth.org.uk/sites/default/files/the_lonely_society_report.pdf
16. Marmot M, Allen J, Goldblatt P, Boyce T, McNeish D, Grady M et al. *Fair Society,*
*Healthy Lives: The Marmot Review*. 2010;
[http://www.instituteofhealthequity.org/resources-reports/fair-society-healthy-lives-the-](http://www.instituteofhealthequity.org/resources-reports/fair-society-healthy-lives-the-marmot-review/fair-society-healthy-lives-full-report-pdf)
[marmot-review/fair-society-healthy-lives-full-report-pdf](http://www.instituteofhealthequity.org/resources-reports/fair-society-healthy-lives-the-marmot-review/fair-society-healthy-lives-full-report-pdf)
17. Paul K, Moser K. Unemployment impairs mental health: Meta-analyses. *J Vocational*
*Behavior* 2009;74:264-282. doi:10.1016/j.jvb.2009.01.001
18. Prior RW. Editorial. *J Applied Arts Health* 2009;1:3–6.
19. Bygren LO, Johansson S, Konlaan BB, Grijbovski AM, Wilkinson AV, Sjöström M.
*Attending cultural events and cancer mortality: A Swedish cohort study. Arts Health: Int J*
*Res Pol Pract* 2009;1:64–73. doi:10.1080/17533010802528058
20. Camic P, Chatterjee H. Museums and art galleries as partners for public health
interventions. *Perspect Public Health* 2013;133:66–77.
21. Chatterjee H. Museums and art galleries as settings for public health interventions. In S
Clift, P Camic (Eds.), *Oxford Textbook of Creative Arts, Health, and Wellbeing:*
*International Perspectives on Practice, Policy and Research*. New York, Oxford: Oxford
University Press 2016; p.281–289.
22. Clift S, Skingley A, Coulton S, Rodriguez J. The effectiveness and cost-effectiveness of a
participative community singing programme as a health promotion initiative for older people: protocol for a randomised controlled trial. *BMC Public Health* 2011;11:1–6. doi:10.1186/1471-2458-11-142

- 23. Clift S, Camic P. Oxford Textbook of Creative Arts, Health, and Wellbeing: International
Perspectives on Practice, Policy and Research. New York, Oxford: Oxford University
Press, 2016.
- 24. Konlaan BB, Bygren LO, Johansson S. Visiting the cinema, concerts, museums or art
exhibitions as determinant of survival: a Swedish fourteen-year cohort follow-up. *Scand J*
*Public Health* 2000;28:174–178. doi:10.1177/14034948000280030501
- 25. Staricoff RL. Arts in health: A review of the medical literature. London: Arts Council
England, Research Report 2004;36:1–91. [http://www.artsandhealth.ie/wp-](http://www.artsandhealth.ie/wp-content/uploads/2011/08/AHReview-of-Medical-Literature1.pdf)
[content/uploads/2011/08/AHReview-of-Medical-Literature1.pdf](http://www.artsandhealth.ie/wp-content/uploads/2011/08/AHReview-of-Medical-Literature1.pdf)
- 26. Staricoff RL. Arts in health: The value of evaluation. *Perspect Public Health*
2006;126:116–120. <http://journals.sagepub.com/doi/pdf/10.1177/1466424006064300>
- 27. Chatterjee HJ, Noble G. Museums, Health and Wellbeing. Farnham: Ashgate, 2013.
- 28. Cuypers K, Krokstad S, Holmen TL, Knudtsen MS, Bygren LO, Holmen J. Patterns of
receptive and creative cultural activities and their association with perceived health,
anxiety, depression and satisfaction with life among adults: the HUNT study, Norway. *J*
*Epidemiol Community Health* 2012;66:698–703.
<https://jech.bmj.com/content/jech/66/8/698.full.pdf>
- 29. Kidd B, Zahir S, Khan S. Arts and Refugees: History, Impact and Future. London: Arts
Council England, 2008. [https://baringfoundation.org.uk/wp-](https://baringfoundation.org.uk/wp-content/uploads/2014/10/ArtsandRefugees.pdf)
[content/uploads/2014/10/ArtsandRefugees.pdf](https://baringfoundation.org.uk/wp-content/uploads/2014/10/ArtsandRefugees.pdf)
- 30. Kemmis S, McTaggart R, Nixon R. The Action Research Planner: Doing Critical
Participatory Action Research. Singapore: Springer 2014.
- 31. Bradbury H. Introduction: How to situate and define action research. In H Bradbury (Ed.).
The Sage Handbook of Action Research, London: Sage 2015. p.1–9.
- 32. Nicolaidis C, Raymaker DM. Community based participatory research with communities
defined by race, ethnicity, and disability: Translating theory to practice. In H Bradbury
(Ed.), The Sage Handbook of Action Research. London: Sage 2015. p.167–178.
- 33. Daykin N, Stickley T. The role of qualitative research in arts and health. In S Clift, P
Camic (Eds.), Oxford Textbook of Creative Arts, Health, and Wellbeing: International
Perspectives on Practice, Policy and Research. New York, Oxford: Oxford University
Press 2016, p.73–82.
- 34. Vaughn LM, Jacquez F. Community-Based Participatory Research Studies Involving
Immigrants. In S Coughlin, SA Smith, ME Fernandez (Eds.), Handbook of Community-

1 Based Participatory Research. New York, Oxford: Oxford University Press, 2017. p.115–
130.
35. Blumenthal DS. Is Community-based participatory research possible? *Am J Prev Med.*
2011;40:386–389.
36. Braun V, Clarke V. Using thematic analysis in psychology. *Qual Res Psychol* 2006;3:77–
101.
37. Pope C, Mays N. *Qualitative Research in Health Care*, 3rd Edition. Oxford: Blackwell,
2006.
38. Pope C, Ziebland S, Mays N. Analysing Qualitative Data. In C Pope, N Mays. *Qualitative*
*Research in Health Care*, 3rd Edition. Oxford: Blackwell. 2006. p.63–81.

[revised manuscript text omitted]

-

43 1

Research and reporting methodology

Revised **Standards for Quality Improvement Reporting Excellence (SQIRE 2.0)**

publication guidelines

Notes to authors

- ▶ The SQIRE guidelines provide a framework for reporting new knowledge about how to improve healthcare.
- ▶ The SQIRE guidelines are intended for reports that describe system level work to improve the quality, safety and value of healthcare, and used methods to establish that observed outcomes were due to the intervention(s).
- ▶ A range of approaches exists for improving healthcare. SQIRE may be adapted for reporting any of these.
- ▶ Authors should consider every SQIRE item, but it may be inappropriate or unnecessary to include every SQIRE element in a particular manuscript.
- ▶ The SQIRE glossary contains definitions of many of the key words in SQIRE.
- ▶ The explanation and elaboration document provides specific examples of well-written SQIRE items and an in-depth explanation of each item.
- ▶ Please cite SQIRE when it is used to write a manuscript.

Text section and item name	Page/line no(s). info is located
Title and abstract	
1. Title	1/1-2/1
Indicate that the manuscript concerns an initiative to improve healthcare (broadly defined to include the quality, safety, effectiveness, patient-centredness, timeliness, cost, efficiency and equity of healthcare).	
Manuscript concerns an initiative to improve healthcare for forcibly displaced people.	
2. Abstract	1/20-2/9
a. Provide adequate information to aid in searching and indexing.	1/21-2/9
b. Summarise all key information from various sections of the text using the abstract format of the intended publication or a structured summary such as: background, local problem, methods, interventions, results, conclusions.	1/21-2/9
The structure abstract summarises all key information under headings of: objective, design, setting, participants, results, and conclusions.	1/21-2/9
Introduction: Why did you start?	
3. Problem description - Nature and significance of the local problem.	2/32-3/8 and 3/28-4/11
4. Available knowledge - Summary of what is currently known about the problem, including relevant previous studies.	3/9-3/27
5. Rationale - Informal or formal frameworks, models, concepts and/or theories used to explain the problem, any reasons or assumptions that were used to develop the intervention(s) and reasons why the intervention(s) was expected to work	2/19-2/31 and 4/12-4/24
6. Specific aims - Purpose of the project and of this report.	4/25-4/33
Methods: What did you do?	
7. Context - Contextual elements considered important at the outset of introducing the intervention(s).	4/17-4/27

8. Intervention(s)
a. Description of the intervention(s) in sufficient detail that others could reproduce it.	5/4-5/6 and 5/9-5/21
b. Specifics of the team involved in the work.	5/24-5/27
9. Study of the intervention(s)
a. Approach chosen for assessing the impact of the intervention(s).	5/27-6/8 6/11-6/23
b. Approach used to establish whether the observed outcomes were due to the intervention(s).	6/23-6/27
10. Measures
a. Measures chosen for studying processes and outcomes of the intervention(s), including rationale for choosing them, their operational definitions and their validity and reliability.	6/27-6/29
b. Description of the approach to the ongoing assessment of contextual elements that contributed to the success, failure, efficiency and cost.	7/2-7/11
c. Methods employed for assessing completeness and accuracy of data.	7/11-7/17
11. Analysis
a. Qualitative and quantitative methods used to draw inferences from the data.	7/2-7/10
b. Methods for understanding variation within the data, including the effects of time as a variable.	6/15-6/17 and 7/10-7/17
12. Ethical considerations - Ethical aspects of implementing and studying the intervention(s) and how they were addressed, including, but not limited to, formal ethics review and potential conflict(s) of interest.	15/24-15/27
Results: What did you find?
13. Results
a. Initial steps of the intervention(s) and their evolution over time (eg, time-line diagram, flow chart or table), including modifications made to the intervention during the project.	7/20-8/28
b. Details of the process measures and outcomes.	8/31-12/7
c. Contextual elements that interacted with the intervention(s).	7/31-7/34 8/25-8/28 9/27-9/28
38	d. Observed associations between outcomes, interventions and relevant contextual elements.	7/22-7/28
e. Unintended consequences such as unexpected benefits, problems, failures or costs associated with the intervention(s).	10/32-11/3
f. Details about missing data.	12/23-12/25 14/13-14/18
Discussion: What does it mean?
14. Summary
12/10-12/16 and
a. Key findings, including relevance to the rationale and specific aims.	12/26-13/20
b. Particular strengths of the project.	12/16-12/25
15. Interpretation
a. Nature of the association between the intervention(s) and the outcomes.	14/24-15/2
b. Comparison of results with findings from other publications.	12/26-12/28 12/31-12/34

	13/3-13/6 13/13-14/1
c. Impact of the project on people and systems.	11/18-11/20
d. Reasons for any differences between observed and anticipated outcomes, including the influence of context.	12/10-12/15
e. Costs and strategic trade-offs, including opportunity costs.	12/4-12/7
16. Limitations	
a. Limits to the generalisability of the work.	14/4-14/13
b. Factors that might have limited internal validity such as confounding, bias or imprecision in the design, methods, measurement or analysis.	14/13-14/18
c. Efforts made to minimise and adjust for limitations.	14/18-14/21
Conclusions	
a. Usefulness of the work.	14/24-14/29
b. Sustainability.	14/29-15/2
c. Potential for spread to other contexts.	15/2-15/5
d. Implications for practice and for further study in the field.	15/5-15/8
e. Suggested next steps.	15/8-15/10
Other information	
18. Funding - Sources of funding that supported this work. Role, if any, of the funding organisation in the design, implementation, interpretation and reporting.	15/19-15/21

Ogrinc G, et al. *BMJ Qual Saf* 2015;0:1–7. doi:10.1136/bmjqs-2015-004411

Downloaded from <http://qualitysafety.bmj.com/> on January 2, 2017